# GLO-Roots: an imaging platform enabling multidimensional characterization of soil-grown root systems

Rubén Rellán-Álvarez[1‡], Guillaume Lobet[2], Heike Lindner[1†], Pierre-Luc Pradier[1†§], Jose Sebastian[1†], Muh-Ching Yee[1], Yu Geng[1,3], Charlotte Trontin[1], Therese LaRue[4], Amanda Schrager-Lavelle[5], Cara H Haney[6], Rita Nieu[7], Julin Maloof[5], John P Vogel[3], José R Dinneny[1*]

[1]Department of Plant Biology, Carnegie Institution for Science, Stanford, United States; [2]PhytoSystems, University of Liège, Liège, Belgium; [3]Department of Energy, Department of Energy Joint Genome Institute, Walnut Creek, United States; [4]Department of Biology, Stanford University, Stanford, United States; [5]Department of Plant Biology, University of California, Davis, Davis, United States; [6]Department of Genetics, Department of Molecular Biology, Massachusetts General Hospital, Harvard Medical School, Boston, United States; [7]Western Regional Research Center, United States Department of Agriculture, Albany, United States

*For correspondence: jdinneny@ carnegiescience.edu

[†]These authors contributed equally to this work

Present address: [‡]Unidad de Genómica Avanzada, Laboratorio Nacional de Genómica para la Biodiversidad, Centro de Investigación y de Estudios Avanzados del Instituto Politécnico Nacional, Irapuato, Mexico; [§]Boyce Thompson Institute for Plant Research, United States Department of Agriculture, Ithaca, United States

Competing interests: The authors declare that no competing interests exist.

**Abstract** Root systems develop different root types that individually sense cues from their local environment and integrate this information with systemic signals. This complex multi-dimensional amalgam of inputs enables continuous adjustment of root growth rates, direction, and metabolic activity that define a dynamic physical network. Current methods for analyzing root biology balance physiological relevance with imaging capability. To bridge this divide, we developed an integrated-imaging system called Growth and Luminescence Observatory for Roots (GLO-Roots) that uses luminescence-based reporters to enable studies of root architecture and gene expression patterns in soil-grown, light-shielded roots. We have developed image analysis algorithms that allow the spatial integration of soil properties, gene expression, and root system architecture traits. We propose GLO-Roots as a system that has great utility in presenting environmental stimuli to roots in ways that evoke natural adaptive responses and in providing tools for studying the multi-dimensional nature of such processes.

## Introduction

Plant roots are three-dimensional assemblies of cells that coordinately monitor and acclimate to soil environmental change by altering physiological and developmental processes through cell-type and organ-specific regulatory mechanisms (*Dinneny et al., 2008*; *Duan et al., 2013*). Soil comprises a complex distribution of particles of different size, composition and physical properties, airspaces, variation in nutrient availability and microbial diversity (*Brady and Weil, 2009*; *Lynch and Wojciechowski, 2015*). These physical, chemical, and biological properties of soil can vary on spatial scales of meters to microns, and on temporal scales ranging from seasonal change to seconds. Root tips monitor this environment through locally and systemically acting sensory mechanisms (*Bao et al., 2014*; *Tabata et al., 2014*).

The architecture of the root system determines the volume of soil where resources can be accessed by the plant (rhizosphere) and is under both environmental and genetic control. Plasticity in growth parameters allows the plant to adjust its form to suit a particular soil. Lateral roots, which usually make

**eLife digest** Most plants absorb water and nutrients from the soil via structures called roots. The shape, size, and structure of a plant's root system can change over its lifetime as the plant responds to changes in their local environment. For example, if water is scarce, a plant may develop a very deep root system that is more efficient at capturing water. Understanding how root systems respond to environmental cues may help us to identify the genes and processes involved.

In this study, Rellán-Álvarez et al. report a new live-imaging platform for analyzing root architecture and its regulation. This platform is called Growth and Luminescence Observatory for Roots (or GLO-Roots for short) and uses 'luminescent' markers that allow growing roots to be visualized when plants are grown in thin, soil-filled, transparent pots. GLO-Roots can track the growth of the plant roots as well as the activity of genes that respond to environmental stress. Rellán-Álvarez et al. developed a software tool called GLO-RIA (GLO-Roots Image Analysis) to analyze the resulting images. GLO-RIA performs several different types of image analysis, including one that detects the position, length, and direction of roots, as well as their shape and depth.

Rellán-Álvarez et al. tested the GLO-Roots techniques in various ways, for example, by analyzing the effects that different conditions have on the growth of the roots of the model plant known as *Arabidopsis thaliana*. Depriving the plants of a nutrient called phosphorous caused the roots to grow more horizontally than when phosphorus is plentiful, presumably to allow the plants to expand their search for phosphate in the upper layers of the soil, where this nutrient is usually more abundant. On the other hand, a shortage of water caused the roots to grow more vertically to access water stored deeper in the soil. GLO-Roots can also be used to measure the water content of soil at different depths and how this influences the architecture of the root.

Further experiments on tomato plants and a grass species called *Brachypodium distachyon* revealed the different architectures of their root systems. Rellán-Álvarez et al. propose that this system could be used to study the roots of other plant species in a variety of environmental conditions. This will provide a more detailed understanding of the ways that different plants adapt in response to changes in their environment.

up the majority of the total root system, often grow at an angle divergent from the gravity vector. This gravity set-point angle is controlled by auxin biosynthesis and signaling and can be regulated by developmental age and root type (*Rosquete et al., 2013*). Recent cloning of the *DRO1* quantitative trait locus demonstrates that natural genetic variation is a powerful tool for uncovering such control mechanisms (*Uga et al., 2013*).

Specific root ideotypes (idealized phenotypes) have been proposed to be optimal for acquisition of water and nitrogen, which are distinct from ideotypes for low phosphorus. Based on computational modeling and field studies, the 'steep, deep, and cheap' ideotype proposed by Lynch and colleagues may provide advantages to the plant for capturing water and elements like nitrogen that are water soluble and therefore tend to move in the soil column with water. This ideotype consists of highly gravitropic, vertically oriented roots that grow deep in the soil column and develop large amounts of aerenchyma, which reduces the overall metabolic cost of the root system (*Lynch and Wojciechowski, 2015*). Other nutrients, like phosphorus, which have limited water solubility and are tightly bound to soil particles, usually accumulate in the top layers of soil and favor root systems that are more highly branched and shallow. The low-phosphorus ideotype effectively increases root exploration at the top layers of soil (*Lynch and Wojciechowski, 2015*). Modeling of root system variables shows that optimum architecture for nitrogen and phosphorus uptake is not the same (*Postma and Lynch, 2014*) and suggests tradeoffs that may affect the evolution of root architecture as a population adapts to a particular environmental niche (*Laliberté et al., 2013*).

Clearly, understanding the architecture of root systems and how environmental conditions alter root developmental programs is important for understanding adaptive mechanisms of plants and for identifying the molecular-genetic basis for different response programs. In addition, root systems have complexity beyond their architecture that needs to be incorporated into our understanding of plant–environment interactions. Primary and lateral roots exhibit different stress response programs in *Arabidopsis* (*Duan et al., 2013*; *Tian et al., 2014*) and may play specialized

roles in water and nutrient uptake. Thus, it is important to develop methods that allow for a multidimensional characterization of the root system that includes growth, signaling, and interactions with other organisms. Furthermore, physiological parameters that affect whole-plant responses to the environment, such as transpiration, are likely integrated into such processes, thus, requiring a more holistic approach to studies of root function.

Based on these considerations, we have developed a new root imaging platform, Growth and Luminescence Observatory for Roots (GLO-Roots), which allows root architecture and gene expression to be studied in soil-grown plants. GLO-Roots is an integrated system composed of custom growth vessels, luminescent reporters, and imaging systems. We use rhizotrons that have soil volumes equivalent to small pots and support growth of *Arabidopsis* from germination to senescence. To visualize roots, we designed plant–codon-optimized luciferase reporters that emit light of different wavelengths. To visualize reporter expression, plants are watered with a dilute luciferin solution and imaged afterwards. We have built a custom luminescence-imaging system that automatically captures images of rhizotrons held vertically. The signal from each reporter is distinguished using band-pass filters held in a motorized filter wheel, which enables automated acquisition of images from plants expressing both structural and environmentally or developmentally responsive reporters. We have also developed GLO-RIA (Growth and Luminescence Observatory Root Image Analysis), an ImageJ (*Schneider et al., 2012*) plugin that allows for automated determination of (among other traits) root system area, convex hull, depth, width, and directionality, a metric which quantifies the angle of root segments with respect to gravity. GLO-RIA is also able to relate root system parameters to local root-associated variables such as reporter expression intensity and soil-moisture content.

Overall GLO-Roots has great utility in presenting environmental stimuli to roots in physiologically relevant ways and provides tools for characterizing responses to such stimuli at the molecular level in whole-adult root systems over broad time scales.

## Results

We have developed an integrated platform for growing, imaging, and analyzing root growth that provides advances in physiological relevance and retains the ability to visualize aspects of root biology beyond structure (*Box 1*).

### The GLO-Roots platform

GLO-Roots is comprised of four parts: (i) growth vessels called rhizotrons that allow plant growth in soil and root imaging; (ii) luminescent reporters that allow various aspects of root biology to be tracked in living plants; (iii) GLO1 (Growth and Luminescence Observatory 1) luminescence-imaging system designed to automatically image rhizotrons; (iv) GLO-RIA, an image analysis suite designed to quantify root systems imaged using GLO-Roots.

### Plant growth system

GLO-Roots utilizes custom-designed growth vessels classically known as rhizotrons, which hold a thin volume of soil between two sheets of polycarbonate plastic. Acrylic spacers provide a 2-mm space in which standard peat-based potting mix is added. Black vinyl sheets protect roots from light and rubber U-channels clamp the rhizotron materials together. Plastic racks hold the rhizotrons vertically and further protect the roots from light. Rhizotrons and rack are placed in a black tub and water is added, to a depth of about 2 cm, at the bottom to maintain moisture in the rhizotrons during plant growth. The volume of soil in the rhizotrons (100 cm$^3$) is similar to small pots commonly used for *Arabidopsis* and supports growth throughout the entire life cycle (*Figure 1A–C* and *Figure 1—figure supplement 1*).

## Box 1.

All resources for GLO-Roots, including the original raw data used in the manuscript, sample images, GLO-RIA user manual, the latest software updates, and the source code, can be found at: https://dinnenylab.wordpress.com/glo-roots/.

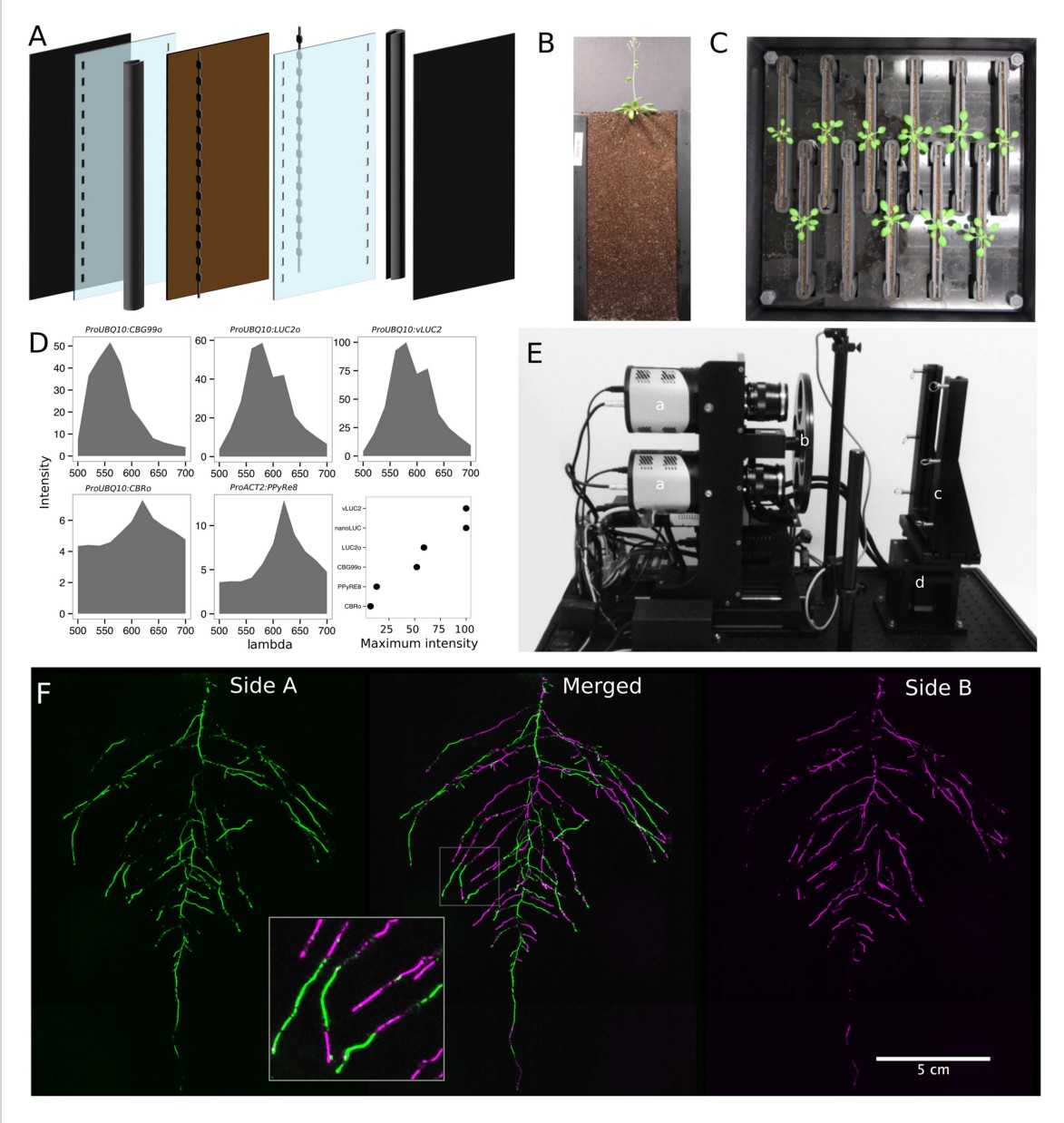

**Figure 1**. GLO-Roots growth and imaging system. (**A**) 3D representation of the different physical components of the rhizotron: plastic covers, polycarbonate sheets, spacers, and rubber U-channels. Blueprints are provided in *Supplementary file 1*. In brown, soil layer. (**B**) A 35-day-old plant in rhizotron with black covers removed. (**C**) Top view of holding box with eleven rhizotrons. (**D**) In vivo emission spectra of different luciferases used in this study. Transgenic homozygous lines expressing the indicated transgenes were grown on agar media for 8 days. Luciferin (300 μM) was sprayed on the seedlings and plates were kept in the dark and then imaged for 2 s at wavelengths ranging from 500 to 700 nm. Five intensity values were taken from different parts of the roots of different seedlings and averaged. Relative maximum intensity values are indicated in the lower right graph. (**E**) GLO1 (Growth and Luminescence Observatory 1)-imaging system. The system is composed of two back illuminated CCD cameras (**a**) cooled down to −55℃. A filter wheel (**b**) allows for spectral separation of the different luciferases. On the right, a rhizotron holder (**c**) is used to position the rhizotrons in front of the cameras. A stepper motor (**d**) rotates the rhizotron 180° to image both sides. (**F**) A 21 DAS plant expressing *ProUBQ10:LUC2o* was imaged on each of two sides of the rhizotron; luminescence signal is colorized in green or magenta to indicate side. In the middle of the panel, a combined image of the two sides is shown. The inset shows a magnified part of the root system.

The following source data and figure supplements are available for figure 1:

**Source data 1**. Two way ANOVA p-values comparing plants grown in MS media vs plants grown in soil (pots or rhizotrons) and plants collected at day or night.

**Source data 2**. Luminescence intensity values of the different luciferase isoforms across the emission spectrum.

*Figure 1. continued on next page*

*Figure 1. Continued*

**Source data 3**. Gene expression values used to construct the PCA of root samples.
**Source data 4**. Gene expression values used to construct the PCA of shoot samples.
**Source data 5**. Shoot Fresh Weight (FW) and primary root length of plants grown with or without luciferin.
**Source data 6**. Ground truth and GLO-RIA measured values of directionality, depth and width use for validation.
**Figure supplement 1**. Effect of different growth systems on gene expression and growth.
**Figure supplement 2**. PCA plot of shoots of the same samples analyzed in *Figure 1*.
**Figure supplement 3**. Image of an Arabidopsis root in soil imaged with white light (brightfield) or epifluorescence.
**Figure supplement 4**. Effect of luciferin addition on primary root length and shoot size of 14 DAS seedlings that were either continuously exposed to 300 µM luciferin from 9 DAS after sowing or not (n = 6-7 plants).
**Figure supplement 5**. GLO-RIA ground truth comparison.

To determine how the biology of plants grown in rhizotrons compares to other standard growth systems, we utilized high-throughput qRT-PCR to study how these conditions affect expression of 77 marker genes in root and shoot samples. These genes were curated from the literature and belong to a wide array of biological pathways including nutrient acquisition, hormone and light response and abiotic stress. Whole root and shoot samples were collected at the end of the light and dark periods (long-day conditions: 16-hr light, 8-hr dark) from plants grown in rhizotrons, pots, and petri dishes with two different media compositions: 1x Murashige and Skoog basal salts (ms) 1% sucrose or 0.25x ms, no sucrose (ms25). Principal component analysis (PCA) of the gene expression values showed a separation of soil- and gel-grown root systems in the first principal components (*Figure 1—figure supplement 1A*, *Figure 1—source data 3*). In roots grown on gel-based media, we observed enhanced expression of genes associated with light-regulated pathways (flavonoid biosynthesis: *FLAVONOL SYNTHASE1, FLS1, CHALCONE SYNTHASE, CHS* and photosynthesis: *RUBISCO SUBUNIT 1A, RBCS1A, CYCLOPHILIN 38, CYP38*), which is expected due to the exposure of gel-grown roots to light. In addition, genes associated with phosphorus nutrition (*LOW PHOSPHATE RESPONSE1, LPR1, PHOSPHATE STARVATION RESPONSE1, PHR1*) were less expressed in soil-grown roots (*Figure 1—figure supplement 1*), suggesting differences in nutrient availability between the different growth systems. Interestingly, shoot samples where not as clearly separated by growth media, and instead, time of day had a greater effect (*Figure 1—figure supplement 2*, *Figure 1—source data 4*). These data suggest root systems may be particularly sensitive to media conditions and indicate that rhizotron-grown root systems more closely approximate the biology of pot-grown plants than standard gel-based media. Shoot weight and primary root length were significantly reduced for gel-grown plants compared to rhizotron- or pot-grown plants suggesting significant differences in the biology of plants grown under these conditions (*Figure 1—figure supplement 1B,C*).

While the 2-mm depth of the soil sheet is 10–20 times the average diameter of an *Arabidopsis* root (between 100 and 200 microns [*Meijon et al., 2013*]), we evaluated whether rhizotron-grown plants exhibited any obvious stress as a consequence of physical constriction. We compared traits of plants growing in vessels that hold similar volumes of soil but in different volumetric shapes (*Figure 1—figure supplement 1*). The number of lateral roots was significantly lower in pot- and cylinder-grown plants compared to rhizotron-grown plants (*Figure 1—figure supplement 1D*), whereas primary root length of rhizotron and cylinder-grown plants was significantly greater than pot-grown plants (*Figure 1—figure supplement 1E*). No significant differences in shoot area were observed between the three systems (*Figure 1—source data 1*). Thus, these data do not support the hypothesis that rhizotron-grown plants experience physical constriction greater than other vessels holding the same volume of soil.

## Generation of transgenic plants expressing different luciferases

*Arabidopsis* roots cannot easily be distinguished from soil using brightfield imaging due to their thinness and translucency (*Figure 1—figure supplement 3*); thus, reporter genes are needed to enhance the contrast between the root and its environment. Luciferase (LUC) is an ideal reporter to visualize roots: (1) unlike fluorescent reporters, luciferase does not require high-intensity excitation light, which could influence root growth, (2) peat-based soil (a type of histosol) exhibits no autoluminescence but does autofluoresce at certain excitation wavelengths similar to GFP (*Figure 1—figure supplement 3*), (3) while GFP is very stable, and thus not well suited for imaging dynamic transcriptional events, the luciferase enzyme is inactivated after catabolism of luciferin, making it ideal for studying dynamic processes such as environmental responses. A considerable number of luciferases have been developed that emit light spanning different regions of the visible spectrum, but their utilization has been limited to studies in animals (*Table 1*).

To determine the efficacy of using luciferase to visualize roots in soil, we codon-optimized sequences of *PpyRE8*, *CBGRed*, *LUC2*, and *CBG99* for *Arabidopsis* expression. In addition, nanoLUC (*Hall et al., 2012*) and venus-LUC2 (*Hara-Miyauchi et al., 2012*) were utilized. Constitutive luciferase expression was driven in plants using the *UBIQUITIN 10 (UBQ10)* or *ACTIN2 (ACT2)* promoters using vectors assembled through a Golden Gate cloning system (*Emami et al., 2013*). Plants homozygous for a single locus T-DNA insertion were evaluated for in vivo emission spectra and luminescence intensity (*Figure 1D*, *Figure 1—source data 2*). All the evaluated luciferases use D-luciferin as a substrate facilitating the simultaneous imaging of different luciferases except nanoLUC, which uses a proprietary substrate furimazine (*Hall et al., 2012*). Luciferases with red-shifted emission spectra were less intense than the green-shifted luciferases (*Figure 1D*). LUC2o showed an emission maximum at 580 nm and a minor peak at 620 nm while CBG99o lacks the minor peak.

Continuous addition of luciferin did not have a significant effect on shoot weight or primary root length (*Figure 1—figure supplement 4*, *Figure 1—source data 5*). After luciferin addition, luminescence signal could be reliably detected in root systems for up to 10 days, depending on the developmental state of the plant.

## GLO1: a semi-automated luminescence-imaging system for rhizotrons

Luminescence-imaging systems commercially available for biomedical research are usually optimized for imaging horizontally held specimens or samples in microtiter plates. Placing rhizotrons in this position would induce a gravitropic response in plants. Working with Bioimaging Solutions (San Diego, CA), we designed and built a luminescence-imaging system optimized for rhizotron-grown plants. GLO1 uses two PIXIS back-illuminated CCD cameras (Princeton Instruments, Trenton, NJ) to capture partially overlapping images of rhizotrons while a motorized stage automatically rotates the rhizotron to capture images of both sides (*Figure 1E*). A composite image is generated from the images captured of each side; *Figure 1F* shows that approximately half of the root system is revealed on each side with few roots being visible on both sides. Apparently, the soil sheet is thick enough to block light from portions of the root system but thin enough to ensure its continuous structure can be compiled from opposite face views. We tested the ability of GLO1-generated images to reveal complete root systems by manually quantifying the number of lateral roots in excavated root systems of eight different

**Table 1**. Luciferases used in this study

| Luciferase | Origin | Maximum wavelength | Substrate |
|---|---|---|---|
| PpyRE8 | Firefly | 618 | D-luciferin |
| CBGRed | Click beetle | 615 | D-luciferin |
| Venus-LUC2 | FP + firefly | 580 | D-luciferin |
| LUC(+) | Firefly | 578 | D-luciferin |
| CBG99 | Click beetle | 537 | D-luciferin |
| Lux operon | *A. fischeri* | 490 | Biosynthesis pathway encoded within operon |
| NanoLUC | Deep sea shrimp | 470 | Furimazine |

plants and testing these results against estimates of lateral root number from images of the same plants visually inspected by four different persons. These comparisons revealed good correlation ($R^2 = 0.974$) between actual lateral root counts and image-based estimation, indicating GLO1-generated root images provide an accurate representation of the in soil root system.

## GLO-RIA: GLO-Roots Image Analysis

We developed a set of image analysis algorithms that were well suited for the complex root systems that GLO-Roots is able to capture. GLO-RIA is an ImageJ plugin divided in two modules.

The first module (RootSystem) performs four different types of analysis: (i) a local analysis that detects all root particles in the image and computes their position, length, and direction; (ii) the global analysis performs a root system level analysis and computes the total visible surface, convex hull, width, and depth; (iii) the shape analysis uses elliptic Fourier descriptors or pseudo-landmarks similarly to RootScape (*Ristova et al., 2013*) to perform a shape analysis on the root system; (iv) the directionality analysis computes the mean direction of root particles in a root system (either on the full image or by a user-defined region of interest in the image). These four analysis methods are fully automated by default, but can be manually adjusted if needed.

The second module of GLO-RIA (RootReporter) was specifically designed for the analysis of multi-layered images such as combinations of gene reporter, root structure, and soil moisture. Shortly, the plugin works as follows: (i) detection of the gene reporters and the structure reporters in their respective images; (ii) if needed, a manual correction can be performed to correct the automated detection; (iii) gene reporters are linked with the soil water content and the structure reporters, based on their proximity; (iv) gene reporter intensity (either absolute or normalized using the structural reporter) is computed; (v) all data are exported and saved to a Root System Markup Language datafile (*Lobet et al., 2015*). Gene and structure reporters can be followed across different time and space points. Using an object-oriented approach, great care has been taken to facilitate the user interactions on the different images to streamline the analysis process. *Table 2* shows a list of root system features extracted using GLO-RIA. GLO-RIA does not currently have the ability to reconstruct the root architecture in itself (topological links between roots). This is a challenge for analyzing images captured by GLO-Roots since soil particles cause disruption of root segments.

We tested the accuracy of the measurements obtained from GLO-RIA using two different ground-truthed data sets. Manual measurement of root system width, depth, and average lateral root angle was determined by hand using ImageJ from an independent set of images corresponding to roots of several *Arabidopsis* accessions growing in control conditions. We also used ArchiSimple (*Pagès et al., 2014*) to generate 1240 images of root system models with contrasting sizes and lateral root angles. Since these images are computationally generated, exact determination of root system parameters was possible. For both ground truth data sets, GLO-RIA quantification provided measurements that were well correlated for all three measured parameters (*Figure 1—figure supplement 5D–F*, *Figure 1—source data 6*). Sample images of real and ArchiSimple generated root images are shown with GLO-RIA-defined directionality color-coding (*Figure 1—figure supplement 5G–I*).

## Continuous imaging of root growth

The size of our rhizotrons enables undisturbed root system development (before roots reach the sides or the bottom of the rhizotron) for about 21–23 days for the Col-0 accession growing under long-day conditions (*Figure 2*, *Figure 2—source data 1*); however, root traits

**Table 2**. List of root system features extracted using GLO-RIA

| Variable | Unit |
| --- | --- |
| Projected area | $cm^2$ |
| Number of visible roots | – |
| Depth | cm |
| Width | cm |
| Convex hull area | $cm^2$ |
| Width | cm |
| Feret | cm |
| Feret angle | ° |
| Circularity | – |
| Roundness | – |
| Solidity | – |
| Center of mass | cm |
| Directionality | ° |
| Euclidean Fourier descriptors | – |
| Pseudo landmarks | – |

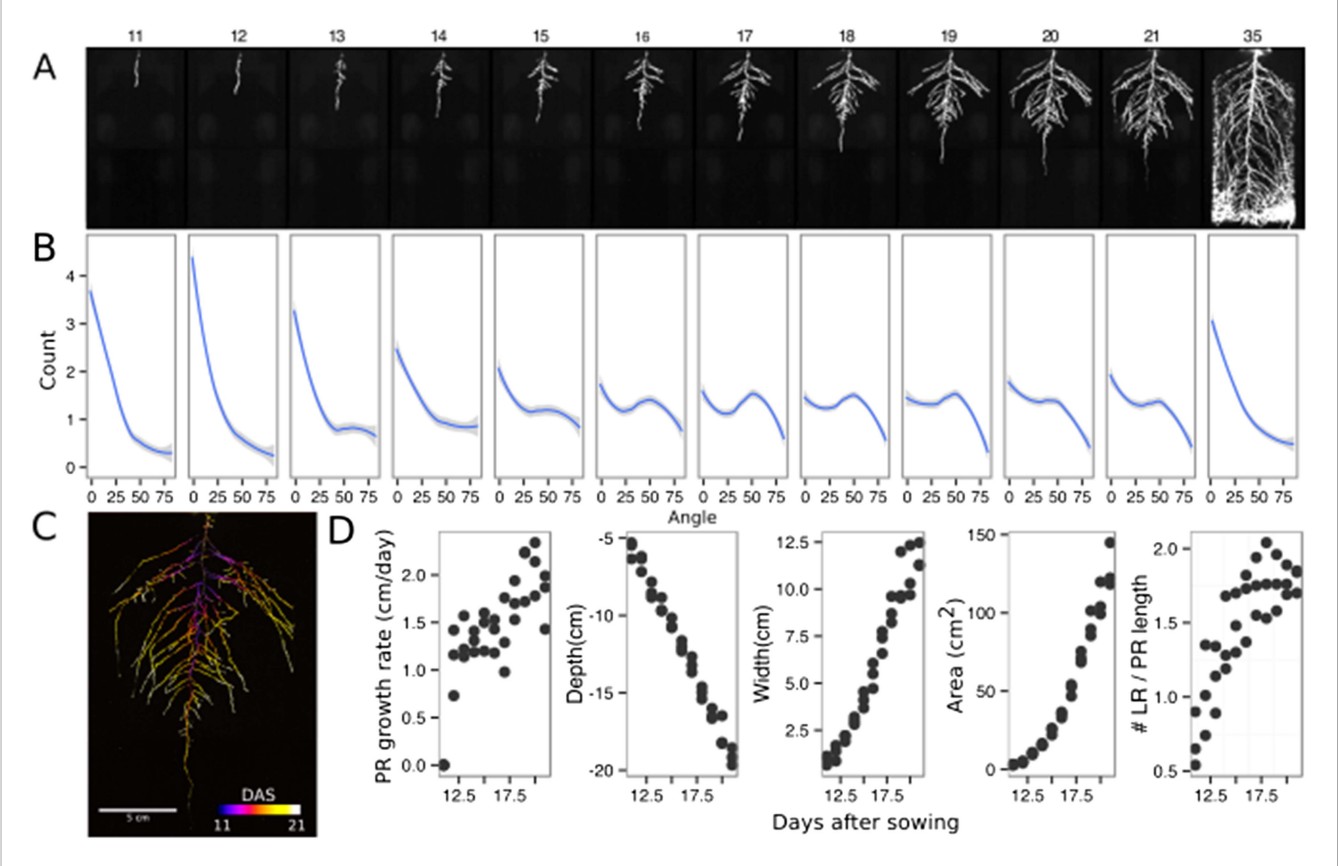

**Figure 2**. Time-lapse imaging of *Arabidopsis* root systems and quantification using GLO-RIA. (**A**) Typical daily time-lapse image series from 11 to 35 DAS of a *ProUBQ10:LUC2o* Col-0 plant. (**B**) Average directionality of three root systems imaged in time series as in panel **A** calculated using the directionality plugin implemented in GLO-RIA. See the GLO-RIA 'Materials and methods' section for information of how the directionality is calculated. (**C**) Color-coded projection of root growth using the images in panel **A**. (**D**) Root system depth, width, root system area are automatically calculated from the convex hull, which is semi-automatically determined with GLO-RIA (n = 3). Primary root length, lateral root number and number of lateral roots divided by the primary root length were quantified manually. A local polynomial regression fitting with 95% confidence interval (gray) was used to represent the directionality distribution curve. 0˚ is the direction of the gravity vector.

The following source data is available for figure 2:

**Source data 1**. Directionality and whole root system architecture trait values from the time series.

such as directionality can be observed through later stages of plant development. See 35 DAS root system and directionality in *Figure 2A,B*. An example of a time series spanning 11 to 21 days after sowing (DAS) of Col-0 roots expressing *ProUBQ10:LUC2o* is shown in *Figure 2A* and *Video 1* with a color-coded time projection shown in *Figure 2C*. Directionality analysis (*Figure 2B*) shows a progressive change in root system angles from 0˚ (vertical) to 55˚ as lateral roots take over as the predominant root type. *Figure 2D* shows the evolution over time of several root traits that can be automatically captured by GLO-RIA (depth, width, area) and others that were manually quantified (primary root growth rate or number of lateral roots per primary root).

## Root system architecture of different *Arabidopsis* accessions

As a proof of concept to estimate the utility of our root-imaging system to phenotype adult root system traits, we transformed a small set of accessions (Bay-0, Col-0, and Sha) with the *ProUBQ10: LUC2o* reporter and quantified root system architecture at 22 DAS (*Figure 3A–C*, *Figure 3—source data 1*). GLO-RIA analysis of these root systems identified several root traits that distinguish Col-0, Bay-0, and Sha. Directionality analysis revealed an abundance of steep-angle regions in the root

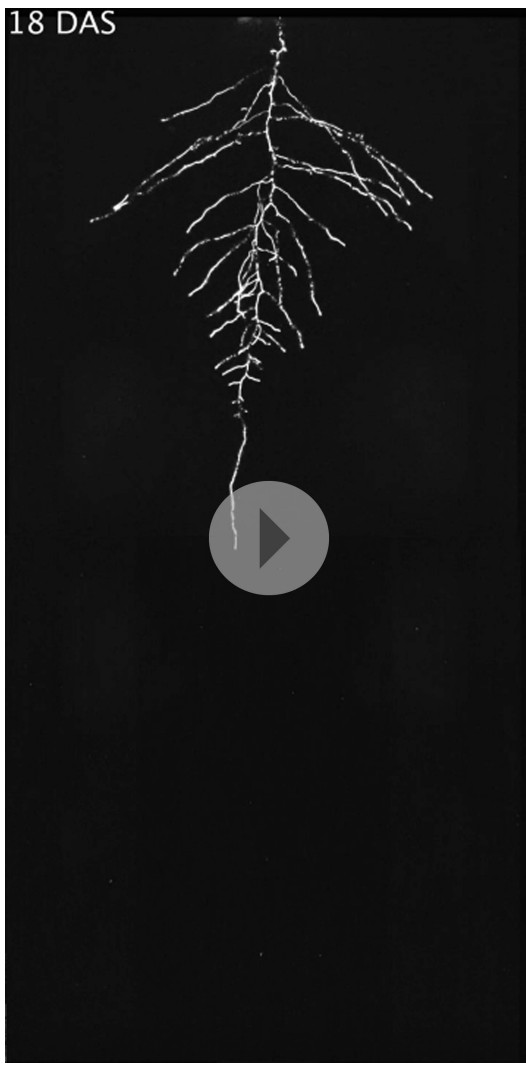

**Video 1.** Time lapse from 11 to 21 DAS of a Col-0 plant expressing *ProUBQ10:LUC2o* grown in control conditions.

system of Bay-0, while Sha showed an abundance of shallow-angled regions and Col-0 was intermediate (*Figure 3D*). Bay-0 shows the deepest and narrowest root system leading to the highest depth/width ratio, while Sha has the widest root system (*Figure 3E*). Other root traits such as root system area and the vertical center of mass also showed significant differences (*Figure 3—figure supplement 1B*). Broad-sense heritability values for depth (96.3), area (92.0), depth/width (97.8), width (95.7), and vertical center of mass (95.0) were all higher than 90%.

To capture the richness of root architecture shape, we used GLO-RIA to extract pseudo-landmarks describing the shape of the root system (see 'Materials and methods' for more details) and performed PCA analysis (*Figure 3—source data 2*). The first principal component captures differences in the distribution of widths along the vertical axis and separates Col-0 and Sha from Bay-0 root systems (*Figure 3F*). Bay-0 shows a homogenous distribution of widths along the vertical axis, while Sha and Col-0 are much wider at the top than bottom. PC2 seems to be capturing a relationship between width at the top and total depth and separates Sha root systems, which are wide at the top and deep from Col-0 root systems, which are wide but not as deep as Sha. Shape information extracted from pseudo-landmarks can distinguish the three different accession using PCA analysis (*Figure 3G*, *Figure 3—source data 3*).

## Spectrally distinct luciferases enable gene expression patterns, characterization of root system interactions, and microbial colonization

We tested whether spectrally distinct luciferase reporters would enable additional information besides root architecture to be captured from root systems. Luciferase reporters have been commonly used to study gene expression and these resources can potentially be utilized to study such regulatory events in soil-grown roots. We transformed *ProACT2:PpyRE8o* into two well-studied LUC reporter lines: the reactive oxygen species response reporter *ProZAT12:LUC* (*Miller et al., 2009*) (*Figure 4A,B*) and the auxin response reporter line *ProDR5:LUC+* (*Moreno-Risueno et al., 2010*) (*Figure 4C,D*). We implemented in GLO-RIA an algorithm that semi-automatically identifies gene reporter signal and associates this object to the corresponding root structure segment. A graphical representation of the results obtained with RootReporter can be observed in *Figure 4—figure supplement 1* (*Figure 4—source data 1*). Reporter intensity values along the first 5 mm of root tips can also be observed in *Figure 4—figure supplement 2* (*Figure 4—source data 2*).

We then took advantage of our ability to constitutively express two spectrally different luciferases and imaged the overlapping root systems (one expressing *ProUBQ10:LUC2o* and the other *ProACT2: PPyRE8o*). While two root systems were distinguishable using this system (*Figure 4—figure supplement 3*); measurements of root system area did not reveal a significant effect on root growth when two plants were grown in the same rhizotron, compared to one; however, further studies are warranted (*Figure 4—figure supplement 3*, *Figure 4—source data 3*).

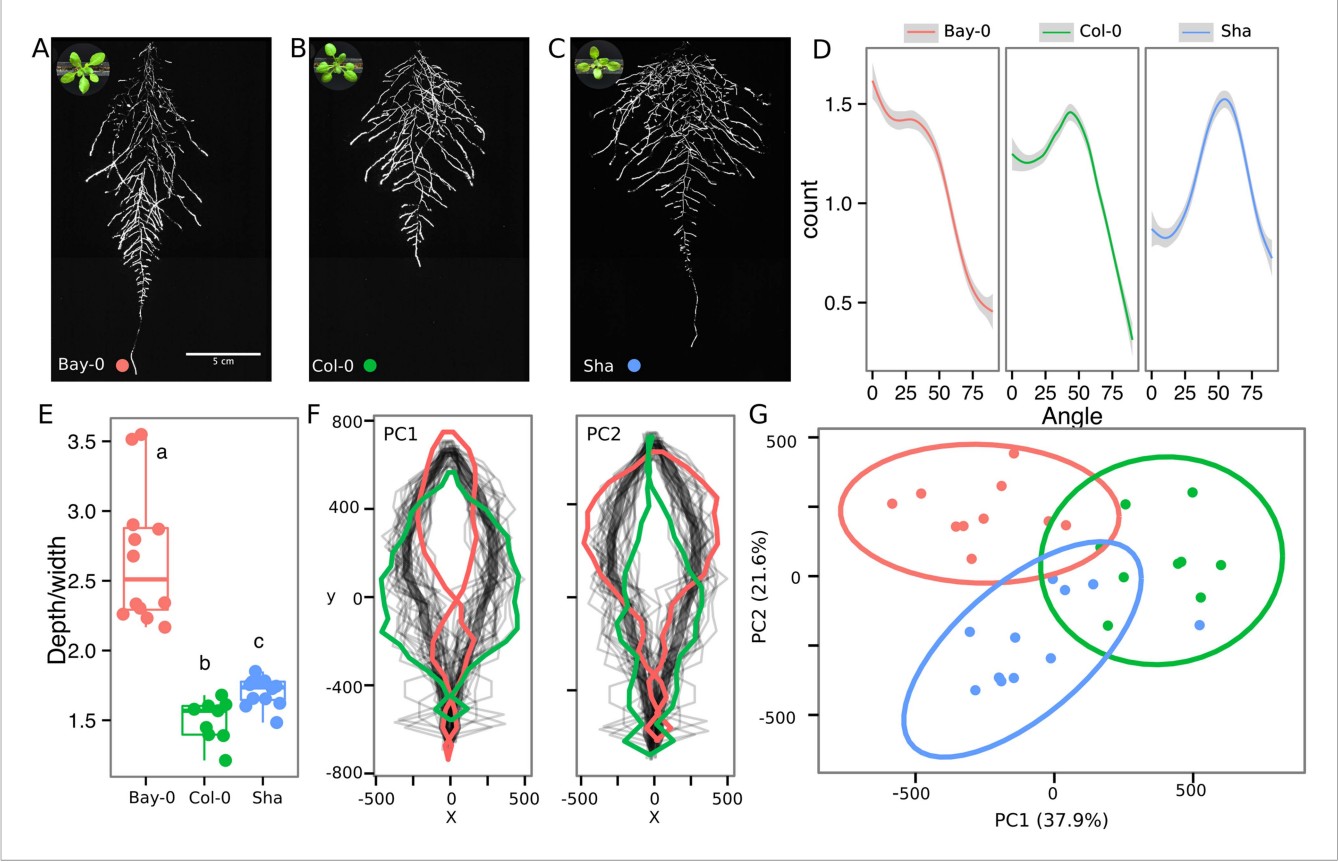

**Figure 3**. Variation in root architecture between accessions of *Arabidopsis*. Representative root and shoot images of (**A**) Bay-0, (**B**) Col-0, and (**C**) Sha accessions transformed with *ProUBQ10:LUC2o* and imaged after 22 DAS. (**D**) Directionality of the root systems, (**E**) depth/width ratio, (**F**) pseudo-landmarks describing shape variation in root system architecture. Eigenvalues derived from the analysis of 9–12 plants per accession are shown. The first two principal components explaining 38% (PC1) and 22% (PC2) of the shape variation are plotted. PC1 captures homogeneity of root system width along the vertical axis and PC2 a combination of depth and width in top parts of the root system. Red and green lines indicate −3SD and +3SD (Standard Deviations), respectively. (**G**) PC separation of the different ecotypes using the PCs described in (**F**). A local polynomial regression fitting with 95% confidence interval (gray) was used to represent the directionality distribution curve. 0° is the direction of the gravity vector. Kolmogorov-Smirnov test at $p < 0.001$ showed significant differences in directionality distributions between all three accessions. Wilcoxon test analysis with $p < 0.01$ was used to test significant differences between the different accessions (n = 9–12 plants).

The following source data and figure supplement are available for figure 3:

**Source data 1**. Directionality, whole root system architectural trait values and shape predictors from Bay-0, Col-0 and Sha.
**Source data 2**. Shape predictor values (TPS format) from Bay-0, Col-0 and Sha used to perform PCA.
**Source data 3**. Whole root system architecture trait values from Bay-0, Col-0 and Sha.
**Figure supplement 1**. (**A**) Root area, (**B**) vertical center of mass of Bay-0, Col-0, and Sha accessions.

The GLO-Roots system uses non-sterile growth conditions, which allows complex biotic interactions that may affect responses to the environment. Bacteria themselves can be engineered to express luminescent reporters through integration of the LUX operon, which results in luminescence in the blue region of the spectrum and is thus compatible with the plant-expressed luciferase isoforms we have tested. *Pseudomonas fluorescens* CH267 (*Haney et al., 2015*), a natural *Arabidopsis* root commensal, was transformed with the bacterial LUX operon and used to inoculate plants. 13 days after inoculation, we were able to observe bacterial luminescence

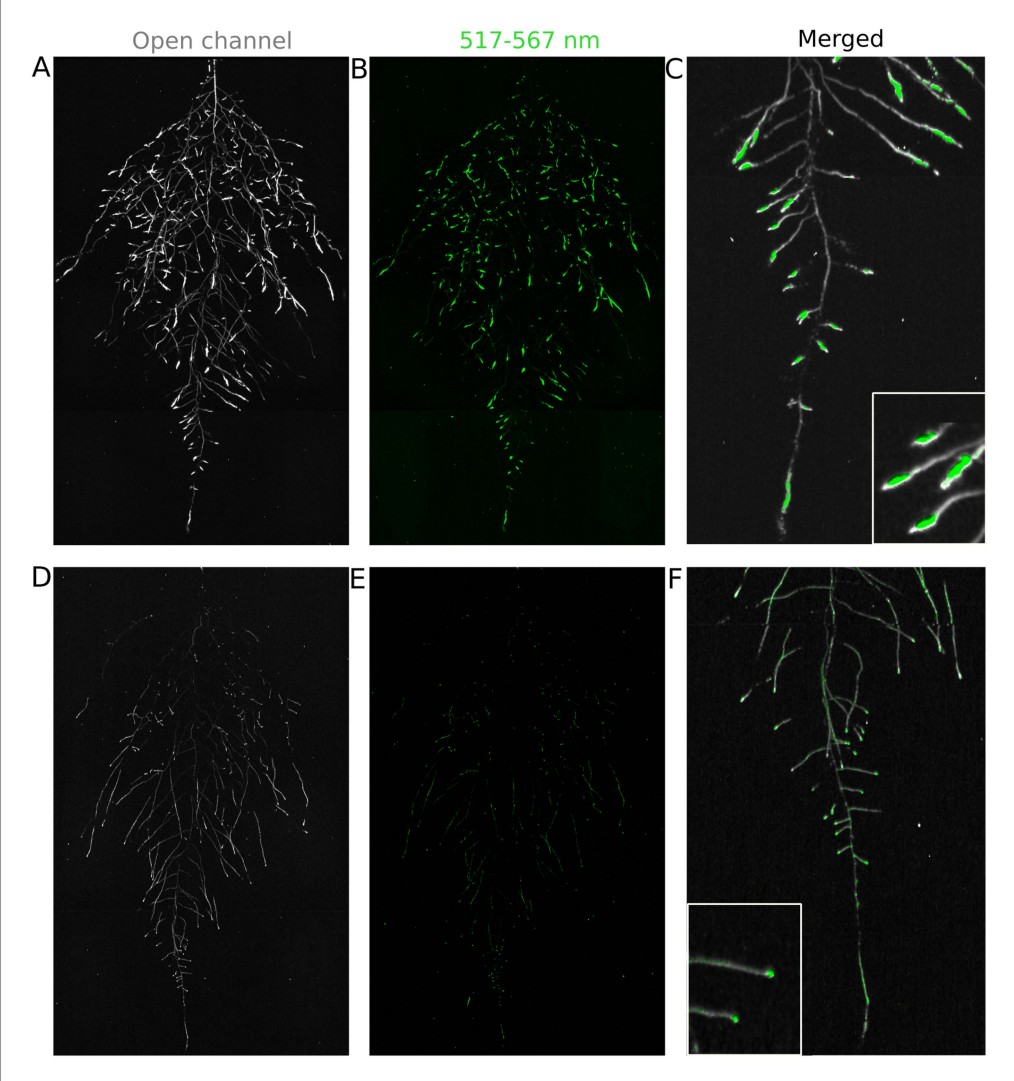

**Figure 4**. Dual-color reporter visualization of root structure and gene expression. Images of whole-root systems (**A**, **D**) or magnified portion of roots (**C**, **F**) at 22 DAS expressing *ProACT2:PPYRE8o* and *ProZAT12:LUC* (green, **A**, **B**, **C**) or *ProDR5rev:LUC*+ (green, **D**, **E**, **F**). Luminescence from PPyRE8 and LUC reporters visualized together using an open filter setting (visualized in grey-scale) while LUC signal is distinguished using a band-pass filter (517 to 567 nm, visualized as green).

The following source data and figure supplements are available for figure 4:

**Source data 1**. Data for *ProZAT12:LUC* reporter gene expression in root segments extracted from a whole root system.

**Source data 2**. Luciferase intensity values from the root tip to maturation zone of *ProUBQ10:LUC2o*, *ProZAT12:LUC* and *ProDR5:LUC*+.

**Source data 3**. Distances to boundary between plants.

**Figure supplement 1**. *ProZAT12:LUC* intensity and root segments automatically identified with GLO-RIA.

**Figure supplement 2**. *ProDR5rev:LUC*+, *ProUBQ10:LUC2o*, and *ProZAT12:LUC* intensity values along the root tip.

**Figure supplement 3**. Images of plants at 22 DAS growing in the same rhizotron and expressing different luciferases.

**Figure supplement 4**. Three reporter-based analysis of root–root–microbe interactions.

colocalizing with plant roots. *P. fluorescens* did not show an obvious pattern of colonization at the root system scale level (*Figure 4—figure supplement 4*). As a proof-of-principle test of the multi-dimensional capabilities of the GLO-Roots system, we visualized both *LUC2o* and *PPyRE8o* reporters in plants and the LUX reporter in bacteria in the same rhizotron (*Figure 4—figure supplement 4*).

## Adaptive changes in root system architecture under water deficit, phosphorus deficiency, and light

To test the utility of the GLO-Roots system to understand response of root systems to environmental stimuli, we tested the effects of light and conditions that mimic drought and nutritional deficiency. To examine the effects of light exposure on the root architecture, the black shields, which normally protect the soil and roots from light, were removed from the top half of the rhizotrons 10 DAS. Using directionality analysis, we detected a significant increase in the steepness of roots only in the light-exposed region of the rhizotron, while the lower shielded region showed no difference (*Figure 6—figure supplement 3A,B* and *Figure 6—figure supplements 4*, *5*). Light can penetrate the top layers of soil (*Mandoli et al., 1990*) and it has been proposed to have a role in directing root growth especially in dry soils (*Galen et al., 2007*) through the blue light receptor *phot1*. Root directionality was not significantly different between light- and dark-treated roots of the *phot1/2* double mutant (*Figure 6—figure supplement 3B*, lower panel, *Figure 6—source data 3*). suggesting that blue light perception is indeed necessary for this response (*Galen et al., 2007*; *Moni et al., 2014*). These data highlight the strong effects of light on root system architecture (*Yokawa et al., 2013*), which GLO-Roots rhizotrons are able to mitigate.

Plants grown in low-P soil showed a significant increase in the width–depth ratio of the root system compared to plants grown in P-replete soil, as determined using the automated root system area finder in GLO-RIA (*Figure 6—figure supplement 2A,B*, *Figure 6—source data 2*). Plants under P deficiency showed an increase in the ratio between root–shoot area (*Figure 6—figure supplement 2C*), which indicates a higher investment of resources in the development of the root system at the expense of shoot growth (*Figure 6—figure supplement 2D*). Root systems of control and P-deficient plants showed no significant differences in directionality at 22 DAS but at 27 DAS, roots were more horizontally oriented in P-deficient plants (*Figure 6—figure supplement 2E*). The observed changes in root architecture are consistent with root system ideotypes that improve phosphorus uptake efficiency.

GLO-Roots is especially well suited for studying water-deficit (WD) responses. First, shoots are exposed to the atmosphere and vapor pressure deficit is maintained at levels that allow for transpiration of water from the shoot. Second, soil in rhizotrons is exposed to air at the top and dries from the top-down; drying soil increases the volume occupied by air and reduces contact of root with liquid water, all of which are similar to changes in soil expected in the field during WD. Finally, as peat-based soil dries, its optical properties change, allowing moisture content to be approximated from brightfield images. We took advantage of the change in gray-scale pixel intensity to construct a calibration curve (*Figure 5—figure supplement 1*, *Figure 5—source data 1*) that quantitatively relates gray-scale pixel intensity to moisture content (*Figure 5A*); water content can be color coded in images with appropriate look-up tables (*Figure 5B*). Soil color was not affected by the presence or absence of roots (*Figure 5—figure supplement 2*). Using this approach, water content in a rhizotron can be mapped and visualized in 2D (*Figure 5C,D*). In the example shown, we can observe that a 22 DAS Bay-0 plant-depleted soil-moisture content locally around the root system (*Figure 5E*).

We performed several trials to simulate WD in our growth system. Plants were germinated, grown under control conditions then transferred to 29°C, and standing water was removed from the container holding the rhizotrons starting at 9 DAS or 13 DAS. Elevated temperature combined with water deficit is a common stress that modern crop varieties are poorly adapted to, thus, highlighting the importance of examining this combined treatment (*Lobell et al., 2014*; *Ort and Long, 2014*). Plants were maintained in this WD regime until 22 DAS when luciferin solution was added and the plants imaged. At 13 DAS, lateral roots near the soil surface are already emerged (*Video 1*, *Figures 2A*) and 9 days of subsequent WD treatment caused lateral roots to show an increase in gravitropism leading to the development of a root system that was deeper and more vertically oriented (*Figure 6A*). Roots of Bay-0 plants showed similar responses, though the extent of change was less

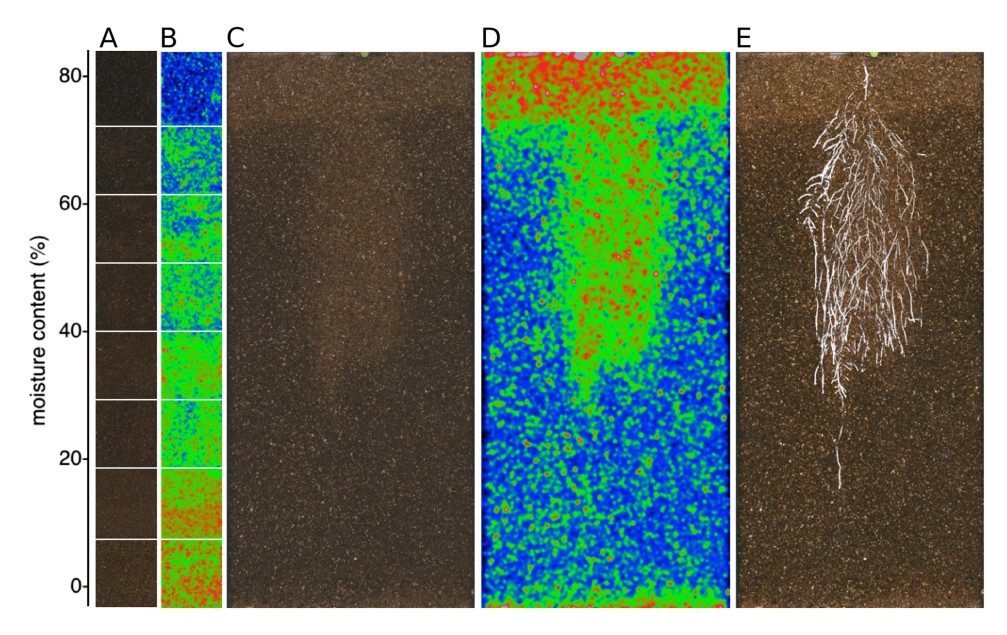

**Figure 5**. Soil moisture and root architecture mapping in rhizotrons. (**A**) Composite image showing regions of soil taken from rhizotrons prepared with different moisture levels. (**B**) Differences in gray-scale intensity values were enhanced using a 16-color look-up table (LUT). Brightfield image of soil in rhizotron (**C**) and converted using 16-color LUT to enhance visualization of distribution of moisture (**D**). (**E**) Root system of a Bay-0 22 DAS subjected to WD since 13 DAS. Root system visualized using luminescence and overlaid on brightfield image of soil in (**C**).

The following source data and figure supplements are available for figure 5:

**Source data 1**. Pixel intensity and water content values used to construct calibration curve.

**Figure supplement 1**. Moisture calibration curve.

**Figure supplement 2**. Comparison of soil intensity values between areas of the rhizotron with or without the presence of roots, determined based on luminescence data.

pronounced since Bay-0 roots are normally more vertically oriented (*Figure 6B*). Plants transferred at 9 DAS and grown for 13 days under WD showed less lateral root development in the top layer of soil (*Figure 6E*). At this time point, lateral roots start to emerge (*Video 1*) and early drought may lead to growth quiescence or senescence. Careful examination of roots in these regions showed evidence of small lateral root primordia populating the primary root (*Figure 6F*). After 24 hr of re-watering (*Figure 6G*), these lateral root primordia reinitiated growth (*Figure 6H*).

Time-lapse imaging of the water-deficit response showed that changes in root growth direction occurred ahead of the dry soil front (*Video 2*). Using GLO-RIA, we were able to correlate local water-moisture contents with the orientation of root segments. With this approach, we observed that root segments in dryer areas of the rhizotron grew at steeper root angles (*Figure 7*, *Figure 7—source data 1*) than roots in wetter regions, though lateral root angle in these regions was also affected. These data suggest that both local and systemic signaling are likely involved in redirecting lateral roots deeper during the simulated drought treatments tested here.

We also grew plants under WD at control temperatures or under WW conditions at elevated temperature to test the effects of these individual stresses on root architecture. We observed that both conditions were sufficient to induce a change in root directionality indicating that the plant uses similar mechanisms to avoid heat and water-deficit-associated stresses (*Figure 6—figure supplement 1*). We next asked which regulatory pathways controlled the observed changes in lateral root

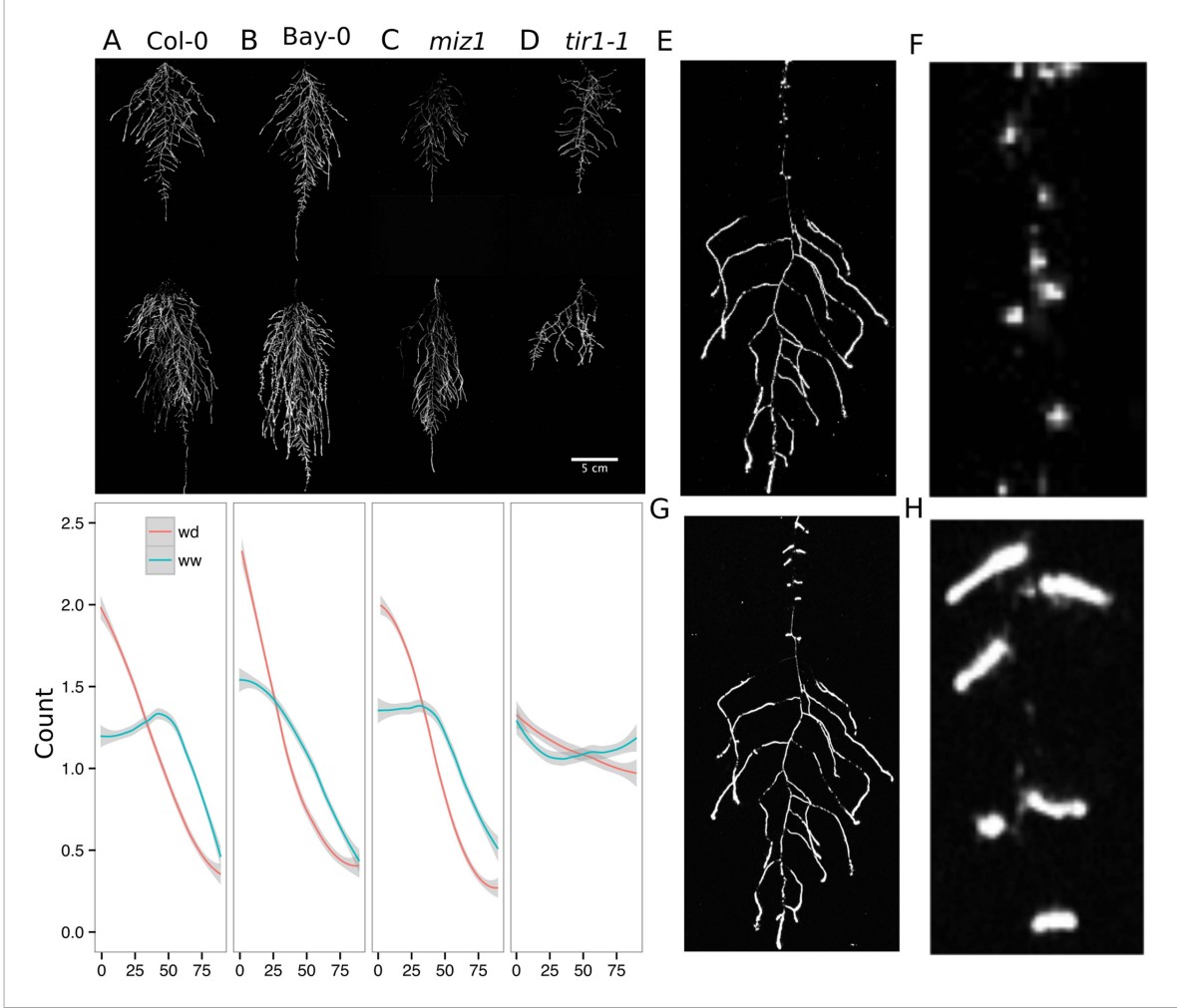

**Figure 6**. Study of effect of water deficit on root system architecture. (**A–D**) Root systems 22 DAS and exposed to water deficit 13 DAS onwards (n = 8-9 plants). Sample images of WW (upper panels) and WD (lower panels) root systems treated from 13 DAS and directionality (line graphs to left of images) for (**A**) Col-0 (**B**) Bay-0 (**C**) *miz1* and (**D**) *tir1-1*. (**E**) Root system of a 22 DAS plant exposed to water deficit from 9 DAS onwards with magnified view of lateral root primordia (**F**). (**G**) The same root as in (**E**) 24 hr after re-watering and magnified view of lateral roots (**H**). Kolmogorov–Smirnov test at p < 0.001 showed significant differences in directionality distributions between the WW and WD conditions for all genotypes except *miz1*. A local polynomial regression fitting with 95% confidence interval (gray) was used to represent the directionality distribution curve. 0° is the direction of the gravity vector.

The following source data and figure supplements are available for figure 6:

**Source data 1**. Directionality values of Bay-0, Col-0, *miz1, tir1-1* grown under WW, WD and high and control temperature conditions.

**Source data 2**. Directionality, root system architecture traits and shoot area values of Col-0 plants grown under different phosphorus concentrations.

**Source data 3**. Directionality values of Col-0 and *phot1/2* plants grown with the root system in the dark or exposed to light in the top third of the rhizotron.

**Source data 4**. Directionality values at different depths of the rhizotron for Col-0 plants exposed to light in the top third of the rhizotron.

**Source data 5**. Relative water content of leaves from plants grown under WW and WD conditions and high or control temperatures.

**Figure supplement 1**. Directionality analysis of roots of plants transferred to WD conditions after 9 DAS and kept 22°C (control temperature) or 29°C (high temperature) until 22 DAS.

directionality during simulated drought. Hydrotropism is a known environmental response that directs root growth towards wet regions of soil. *MIZ1* is an essential regulator of hydrotropism (*Kobayashi et al., 2007*); however, *miz1* mutants had no significant effect on WD-induced changes in root directionality, compared to wild type (*Figure 6C*, *Figure 6—source data 1*), indicating that this response was distinct from hydrotropism. Auxin is an important mediator of gravitropism and auxin treatment causes lateral roots to grow more vertically (*Rosquete et al., 2013*). Consistent with this role for auxin, mutant plants with loss of function in the auxin receptor TIR1 (*Dharmasiri et al., 2005*, *Kepinski and Leyser, 2005*) showed a near-random distribution of root angles and did not show changes in the root system directionality between WW and WD conditions (*Figure 6D*).

## GLO-Roots for *Brachypodium* and tomato

To examine the general applicability of the GLO-Roots system for other species, we introduced LUC2o-expressing reporters into the model grass *Brachypodium distachyon* and the crop plant *Lycopersicon esculentum* (tomato). *Brachypodium* is well suited to the GLO-Root system because, like *Arabidopsis*, its small size allows mature root systems to be studied in relatively small soil volumes (*Watt et al., 2009*; *Pacheco-Villalobos and Hardtke, 2012*). *LUC2o* driven by the *ZmUb1* promoter was introduced into *Brachypodium* using the pANIC vector (*Mann et al., 2012*). *Brachypodium* roots showed a distinct architecture from *Arabidopsis* marked by prolific development of secondary and tertiary lateral roots (*Figure 8A*). This is consistent with other studies that show that *Brachypodium* has a typical grass root system (*Watt et al., 2009*). Comparison of root system development in rhizotrons with gel-based media showed that root growth is higher in soil than in plates (*Figure 8—figure supplement 1*, *Figure 8—source data 1*). Previous work has suggested that auxin levels in *Brachypodium* roots are sub-optimal for growth (*Pacheco-Villalobos et al., 2013*). Pacheco-Villalobos and colleagues suggest that, in *Brachypodium*, and contrary to what happens in *Arabidopsis*, ethylene represses *YUCCA* reducing the synthesis of auxin. The reduced growth that we observe in plates and the high levels of ethylene that build up in sealed plates (*Buer et al., 2003*) would support this mechanism.

Tomato plants were transformed with *Pro35S:PPyRE8o* and *ProeDR5rev:LUC2* reporters. The plants showed more rapid growth than *Arabidopsis* or *Brachypodium* and required fertilizer to prevent obvious signs of stress (reduced growth, anthocyanin accumulation).

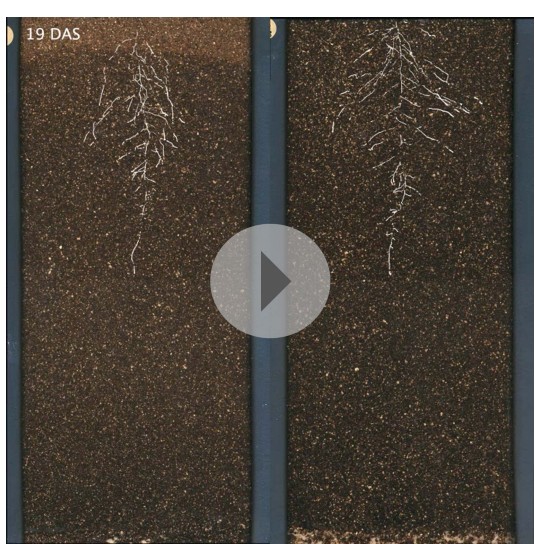

**Video 2.** Time lapse from 16 to 24 DAS of Col-0 plants expressing *ProUBQ10:LUC2o* growing in water-deficient (left) and control (right) conditions. Plants were sown under control conditions and water-deficit treatment started 11 DAS. Images were taken every day.

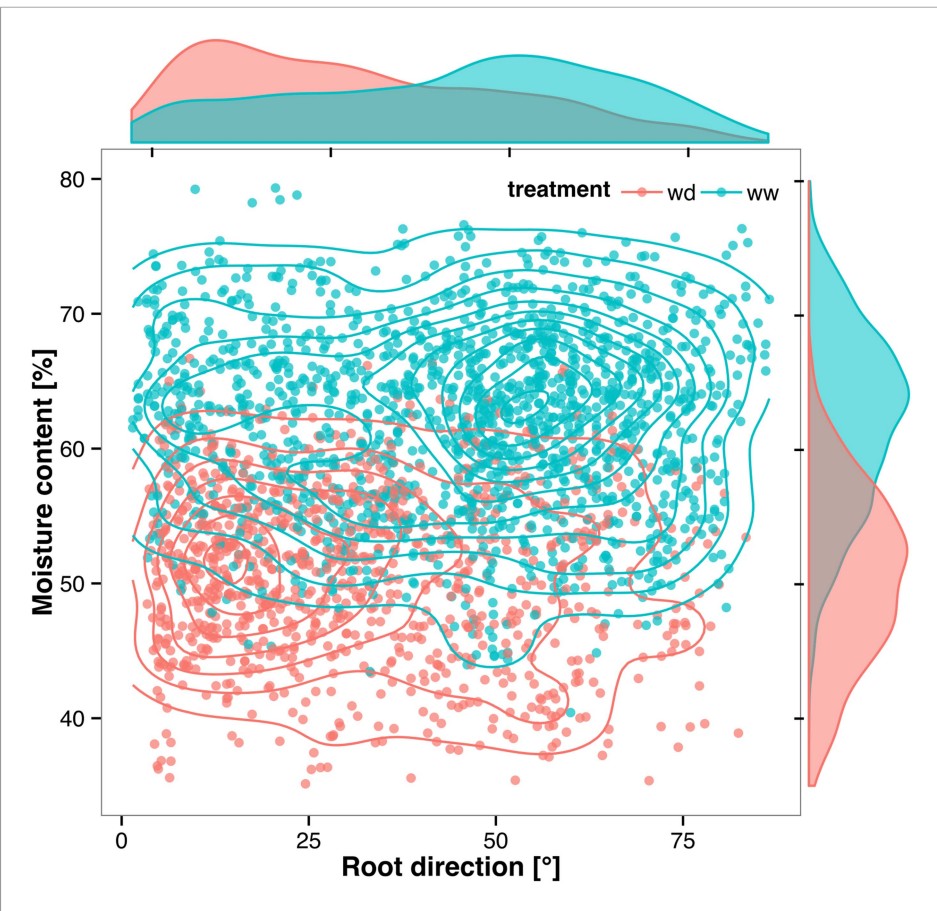

**Figure 7**. Relationship between local soil moisture content and root growth direction. Data quantified from the time-lapse series are shown in **Video 2**. Density plots shown at periphery of graph for root direction (x-axis) and soil moisture (y-axis). 0° is the direction of the gravity vector. Data represent 2535 root tips measured in a series encompassing 10 time points.
The following source data is available for figure 7:

**Source data 1**. Individual root segment traits of plants growing under WW and WD conditions.

Root systems were imaged from 17 DAS plants. Roots showed presumptive lateral root primordia marked by DR5-expression (*Figure 8C,D*). These results show that the GLO-Roots method can be applied to study root systems of different plant species and will likely be useful for studying root systems of other small- to medium-sized model plants and for early stages of larger crop plants.

## Discussion

### GLO-Roots enables a multi-dimensional understanding of root biology

Recent studies of root systems has emphasized structural attributes as important contributors of root system function. Indeed, studies examining the role of genetic variants in tolerating abiotic stress have demonstrated the importance of such characteristics (*Uga et al., 2013*). Roots, however, are highly diverse in the biology they perform and a multi-dimensional understanding of root systems, which incorporates differences in signaling, metabolism, and microbial association as well as structure, may provide a clearer understanding of the degree to which sub-functionalization of the root system plays a role in important processes such as acclimation and efficient resource acquisition.

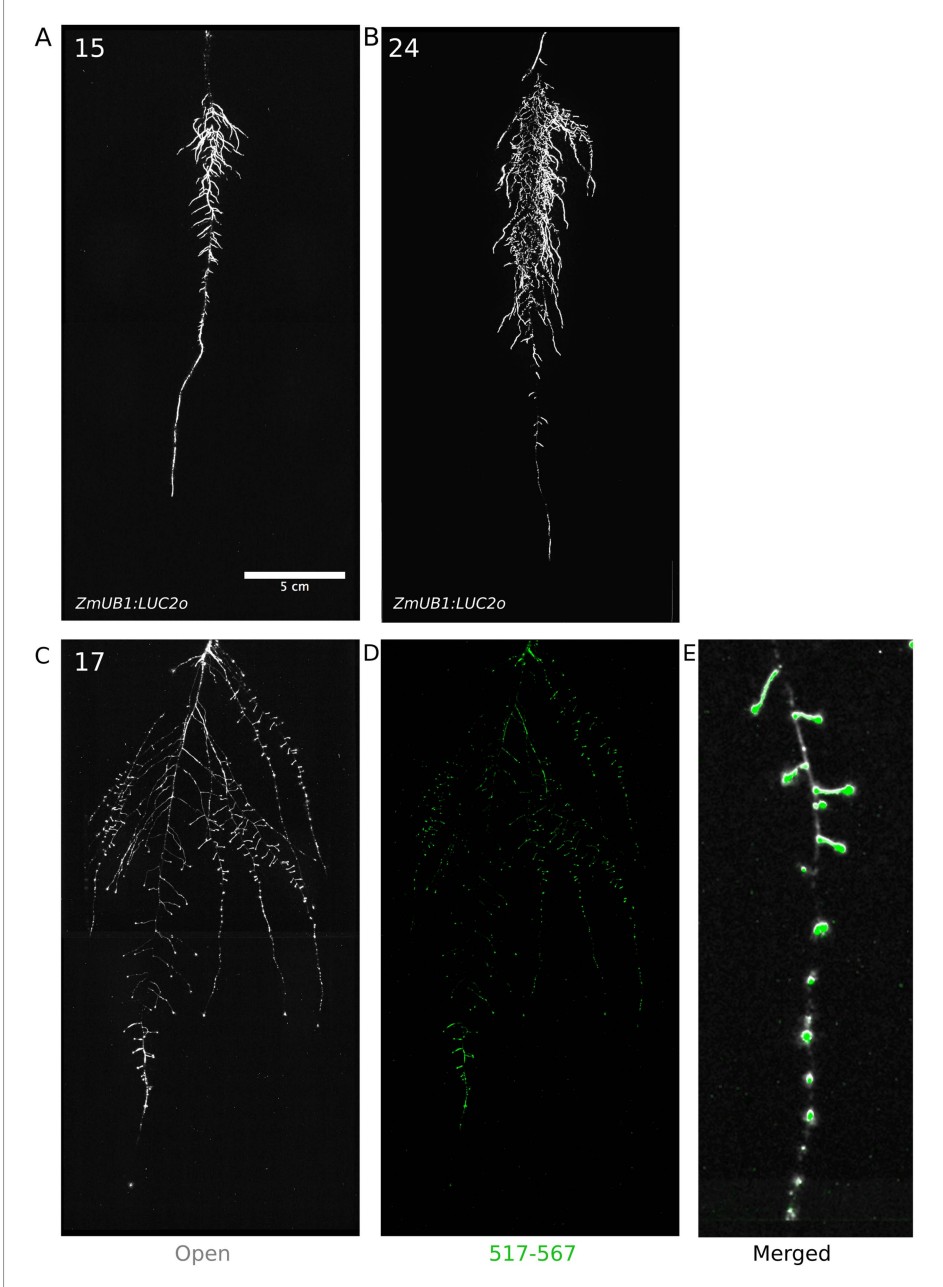

**Figure 8**. Roots of *Brachypodium distachyon* transformed with *ProZmUB1:LUC2o* and imaged at 15 (**A**) and 24 (**B**) DAS grown in control conditions. (**C**) Open channel of 17 DAS tomato plant transformed with *ProeDR5rev:LUC2o* and *Pro35S:PPyRE8o*. (**D**) Green channel showing only *ProeDR5rev:LUC2o*. (**E**) Amplification of the open and green channel showing increased expression of *ProeDR5rev:LUC2o* reporter in early-stage lateral roots.

The following source data and figure supplement are available for figure 8:

**Source data 1**. Depth of *Brachypodium* primary roots grown in petri plates and rhizotrons.

**Figure supplement 1**. Depth of the primary root of Brachypodium plants grown in rhizotrons or on gel-based media (n = 8–11).

We have developed tools in GLO-Roots that allow for tracking multiple aspects of soil physicochemical properties and root biology simultaneously. Using GLO-Roots, we are able to map in 2D coordinates the soil physical properties such as soil moisture together with root architecture traits such as directionality, growth rates, and gene expression levels. All this information is aggregated in layers for each x, y coordinate. Using GLO-RIA, we integrate this multilayer information, leveraging our ability to simultaneously and seamlessly investigate root responses to environmental stimuli such as soil-moisture content. Luciferases that emit light at different wavelengths allow for constitutive and regulated promoters to be studied together. Introduction of luciferase reporters into microbes provides an additional layer of information that is a readout of the association between organisms and how this might be affected by environmental conditions. The flexibility of the GLO-Roots system may enable additional dimensionality to our understanding of root biology. Other physical properties such as $CO_2$ or pH mapping in rhizotrons have already been enabled by using planar optodes (*Blossfeld et al., 2013*). It may be possible to engineer LUX-based reporters in microbes that are responsive to extracellular metabolites, creating microbial biosensors, and integration of such tools may enable root-exudation and nutrition to be analyzed in soil. Split-luciferase reporters have been engineered that allow bi-molecular interactions to be studied. Finally, molecular sensors analogous to Förster resonance energy transfer (FRET) sensors, termed bioluminescence resonance energy transfer (BRET)-sensors (*Shaw and Ehrhardt, 2013*), may allow metabolite tracking dynamically through the root system. With additional innovation in the development of luciferase reporters, the GLO-Roots system will likely expand the repertoire of biological processes that can be studied over an expanded range of developmental time points and environmental conditions.

## Enhanced root growth and gravitropism may constitute an avoidance mechanism used during water-deficit stress

It has been proposed that plants with steep root systems will be better able to tap into deep water resources and thus perform better under water deficit. For example in rice, the IR64 paddy cultivar shows shallow root systems in upland fields, whereas Kinandang Patong, an upland cultivar, is deeper rooting (*Uga et al., 2013*). Plants maintain a number of regulatory pathways that mediate changes in physiology during WD. Enhanced growth of root systems has been well characterized in field-grown plants; however, this has not been recapitulated in studies of gel-grown *Arabidopsis* plants. Thus, it has been unclear whether *Arabidopsis* simply responds to WD differently. Our results here show that *Arabidopsis* does indeed maintain a classical WD response that expands the root system and directs growth downward. Interestingly, under our stress regime, we did not observe a significant decrease in the relative water content of shoot tissues (*Figure 6—figure supplement 5*, *Figure 6—source data 5*), suggesting that the changes in root architecture were sufficient to provide access to deep water and prevent dehydration. Such changes in root growth are likely regulated through systemic and local signaling that involve auxin signaling but acts independently of known pathways that control moisture-directed root growth.

## Perspectives and conclusions

Understanding plant biology requires a sophisticated understanding of how environmental stimuli affect the form and function of plants as well as an understanding of how physiological context informs such responses. Environmental conditions are at least as complex as the plants they affect. Plant roots are exposed to a variety of environmental signals that change in time and space at very different scales that are integrated at the whole-plant system. It is an important challenge in biology to develop methods of growing and studying plants that present such stimuli in a manner that the plant is likely to encounter in nature. After all, the plants we study have evolved to survive through mechanisms that have been selected, over evolutionary time, in nature. It will be interesting for future studies to determine how other environmental stimuli affect root growth using GLO-Roots and whether these responses differ between accessions of *Arabidopsis*. Identification of the genetic loci responsible for phenotypic variation in adult root phenotypes may identify the molecular basis for adaptive variation that exists in this species and potentially identify loci that are useful for breeding efforts needed for the next green revolution.

# Materials and methods

## Growth system

### Rhizotrons and growth system fabrication

Rhizotrons are composed of two sheets of 1/8″ abrasion resistant polycarbonate plastic (Makrolon AR (R)) cut to size using a water jet (AquaJet LLC, Salem, OR), two acrylic spacers cut using a laser (Stanford Product Realization Lab), two rubber U-channels cut to strips 30-cm long (McMaster Carr Elmhurst, IL, part # 8507K33), and two sheets of black 0.030″-thick polypropylene sheets (McMaster Carr part # 1451T21) cut with a straight-edge razor blade. Rhizotron designs were drafted in Adobe Illustrator (Adobe, San José, CA). The blueprints of all the parts are provided in *Supplementary file 1*. The top edge of each polycarbonate sheet was painted with black 270 Stiletto nail polish (Revlon, New York, NY).

### Boxes and holders

Rhizotrons are held vertical during plant growth in a custom rack system composed of two sheets of 1/4″ black acrylic plastic cut with slots for eleven rhizotrons using a laser, four 3/8″-poly(vinyl chloride) (PVC) rods (McMaster Carr part # 98,871a041) secured with PVC nuts (McMaster Carr part # 94,806a031) to hold the acrylic sheets horizontal. The rack is placed inside a 12″ × 12″ × 12″ black polyethylene tank (Plastic Mart part # R121212A).

### Rhizotron preparation

The procedure to construct a rhizotron with soil is as follows: two pieces of polycarbonate plastic are laid flat on a table with the spacers inserted. Using an electric paint gun, a fine mist of water is applied to the bare polycarbonate sheets. Then, using a 2-mm sieve (US Standard Sieve Series N° 10) a fine layer of PRO-MIX(r) PGX soil (Premier Tech, Canada) is applied. Excess soil is discarded by gently tapping the plastic against the table in a vertical position. Water is sprayed again onto the soil, then a second layer of Pro-MIX is applied as before. For P deficiency experiments, soil supplemented with 1 ml of 100 μM P-Alumina (control) and 0-P-Alumina (P deficient) was used. To prevent the soil from falling out of the bottom opening, a 3 × 6 cm piece of nylon mesh or paper towel is rolled into a 1-cm wide tube and placed at the bottom side of the rhizotron. The spacers are removed and replaced by clean spacers. The two faces of the rhizotron are carefully joined together and two rubber U-channels slipped on to clamp all pieces together. Assembled rhizotrons are placed into the rack inside the box and 500 ml of water is added to the box.

### Plant growth

*Arabidopsis thaliana* seeds were stratified for 2 day at 4°C in Eppendorf tubes with distilled water. Seeds were suspended in 0.1% agar and 5 to 10 were sown using a transfer pipette in the rhizotron. A transparent acrylic sheet was mounted on top of the box and sealed with tape to ensure high humidity conditions that enable *Arabidopsis* germination. 3 days after sowing, the cover was unsealed to decrease humidity and allow the seedlings to acclimate to a dryer environment. From 3 days after sowing (DAS) to the time the first true leaves emerged, it was critical to ensure that the top part of the rhizotron remained humid for proper germination of the plants. Between three and five DAS, the rhizotrons were thinned leaving only the number plants required for that experiment, typically one, except for experiments examining root–root interactions. Unless otherwise stated, all the experiments presented here, treatments were started 10 DAS. Plants were grown under long-day conditions (16-hr light/8-hr dark) using 20–22°C (day/night) and 150 μE m$^{-1}$ s$^{-1}$. Two types of growth environments were used for experiments: a walk-in growth chamber with fluorescent lightning and a growth cabinet with white LED lights. Relative water content measurements were done as previously described (*Barr and Weatherley, 1962*). Tomato seeds were germinated on filter paper and placed in rhizotrons. Peters fertilizer was added to tomato plants during normal watering.

## qRT-PCR analysis

Seeds were surface sterilized as described before (*Duan et al., 2013*) and grown in rhizotrons, 100 cm$^3$ pots, or on two types of 1% agar (Difco, Becton, Dickinson and Company, Franklin Lakes, NJ) media containing either 1× MS nutrients (Caisson Labs, Smithfield, UT) and 1% Sucrose, (termed ms media) or ¼× MS

nutrients only (termed ms25 media). Both media were buffered using 0.5 g/l MES and pH was adjusted to 5.7 with KOH. All plants were grown together in a growth cabinet with LED lights under long-day conditions (16-hr day/8-hr night). Root and shoot tissue was collected separately from individual plants at the end of the day (1 hr before the lights shut off) and at the end of the night (1 hr before lights came on). Three biological replicates were collected for each condition. RNA was extracted using the Plant RNA MiniPrepTM kit (ZYMO Research, Irvine, CA) according to manufacturer's instructions with on-column DNase treatment (Qiagen). cDNA was made using the iScript Advanced cDNA Synthesis for RT-qPCR kit (Bio-Rad) from 200 ng of total RNA. qRT-PCR was performed using a Fluidigm BioMarkTM 96.96 Dynamic Array IFC with the EvaGreen (Bio-Rad) fluorescence probe according to the Fluidigm Advanced Development Protocol number 37. For the analysis, all the reactions with no amplification (Ct = 999) were set to the maximal Ct for that assay type. The two technical replicates were then averaged and dCt values calculated using AT3G07480, AT4G37830, At1g13320, and At1g13440 as reference internal controls. PCA plots were generated with Devium Web (*Dmitry Grapov, 2014*) using dCt values. dCT values were calculated as dCT = CT~gene interest~ − mean(CT~reference gene~). Primers used are listed in *Supplementary file 2*.

## Biological components

### Codon optimization of luciferases
The following luciferases that emit light at different wavelengths were codon optimized for *Arabidopsis* (Genscript, Piscataway, NJ): LUC2: a yellow improved version (Promega, Madison, WI) of the original *Photinus pyralis* (firefly) LUC.

- PpyRE8: a red variant (*Branchini et al., 2010*) of the *P. pyralis* thermostable variant Ppy RE-TS (*Branchini et al., 2007*).
- CBG99: a green variant (Promega, Madison, WI) from yellow click beetle (*Pyrophorus plagiophthalamus*) luciferases.
- CBR: a red variant (Promega, Madison, WI) from yellow click beetle.

### Non-optimized luciferases
We also used the following non-optimized luciferases:

- nanoLUC: a blue luciferase isolated from a deep sea shrimp (*Hall et al., 2012*).
- venusLUC2: a venus-LUC2 fusion reported to show higher luminescence output than LUC2 (*Hara-Miyauchi et al., 2012*).
- A transposon containing the bacterial luciferase-containing LUX operon was integrated into the *P. fluorescens* CH267 (*Haney et al., 2015*) genome by conjugation with *Escherichia coli SM10pir* containing pUT-EM7-LUX (*Lane et al., 2007*) and used to track root microbe colonization. For inoculation, 9 DAS plants were inoculated with 2 ml of an overnight bacterial culture resuspended in 10 mM MgSO$_4$ and diluted to 0.01 OD.

### Generation of single-reporter transgenic plants
We generated transcriptional fusions of all luciferases to constitutive promoters to examine the activity level and emission spectrum of each isoform. The *attL1-attL2* entry clones containing plant–codon-optimized coding sequence of *LUC2*, *PpyRe8*, *CBG99*, and *CBR* were synthesized by Genscript. A DNA fragment including the *UBQ10* promoter region and first intron was amplified from Col–0 genomic DNA with primers incorporating the attB1, attB4 combination sites at the 5′ and 3′, respectively. The PCR product was then introduced into pDONR P4-P1R (Invitrogen, Grand Island, NY) through a classic Gateway BP-reaction. The resulting plasmid, the *attL1-attL2* entry clones with luciferase sequences, an empty *att*R2-attL3* entry clone and the destination vector dpGreenmCherry (*Duan et al., 2013*) were used to construct *ProUBQ10*:LUC2o, *ProUBQ10:PpyRE8o*, *ProUBQ10: CBG99o*, and *ProUBQ10:CBRo* through Gateway LR reactions. The destination vector *dpGreenm-Cherry* contains a plasma membrane-localized mCherry coding sequence driven by the 35S promoter and is used as a selectable marker of transformation at the mature seed stage (*Duan et al., 2013*). We used Golden Gate cloning and the destination vectors that we had generated before (*Emami et al., 2013*) for the following fusions: *ProUBQ10:nanoLUC2*, *ProUBQ10*:venusLUC, *ProACT2:PpyRE8o*. Briefly, the different components of each construct were PCR amplified with complementary BsaI or SapI cutting sites, mixed with the destination vector in a single tube, digested with either BsaI or SapI, ligated with T4 DNA ligase, then transformed into *E. coli* Top10 cells and plated on LB antibiotic

plates containing X-gal as previously described (*Emami et al., 2013*). Junction sites were confirmed by sequencing. We used pSE7 (Addgene ID #: pGoldenGate-SE7: 47676) as the destination vector of the *ProUBQ10:nanoLUC2*, *ProUBQ10:venusLUC* constructs and pMYC2 (Addgene ID #: pGoldenGate-MCY2: 47679) as the destination vector for *ProACT2:PpyRE8o*. Maps of all the vectors can be found in *Supplementary file 3*. *ProUBQ10:LUC2o* was transformed into Col-0, Bay, and Sha accessions, the *tir1-1* (*Ruegger et al, 1998*) mutant and the *miz1* (*Moriwaki et al., 2011*) T-DNA insertion line (SALK_126928).

### B. distachyon

The *Arabidopsis* plant–codon-optimized Luciferase gene, *LUC2o*, was inserted into the monocot vector pANIC10 via Gateway cloning (*Mann et al., 2012*). *B. distachyon* plants were transformed using the method of Vogel and Hill (*Vogel and Hill, 2008*).

### Tomato

The transcriptional fusion *ProeDR5:LUC2* was generated by cloning the *ProeDR5:LUC2* DNA fragment into the pBIB expression vector via restriction sites SalI and Acc65I. The eDR5 promoter is an enhanced version of DR5 containing 13 repeats of the 11-nucleotide core DR5 element (*Covington and Harmer, 2007*) and the pBIB expression vector contains an NPTII resistance gene under the control of the NOS promoter for use as a selectable marker during transformation into cultivar M82, accession LA3475. All tomato transformations were performed by the Ralph M. Parsons Foundation Plant Transformation Facility (University of California, Davis).

### Generation of dual-reporter plants

To generate dual-reporter plants expressing luciferase isoforms that emit light with divergent emission spectra, we used *ProACT2:PpyRE8o* as the root structural marker and *ProZAT12:LUC* (*Miller et al., 2009*) and *ProDR5:LUC+* (*Moreno-Risueno et al., 2010*) lines that were transformed with the *ProACT2:PpyRE8o* construct. All constructs were transformed using a modified floral dip method as described in *Duan et al. (2013)*.

To make the dual color tomato plants, the *Pro35S:PpyRE8o* transcriptional fusion was generated by putting the plant–codon-optimized coding sequence described above into the pMDC32 expression vector through a Gateway LR reaction. The pMDC32 vector contains a hygromycin resistance gene under the control of the 35S promoter for use as a selectable marker during transformation. This construct was transformed into the transgenic *ProeDR5:LUC2* tomato line.

### In vivo emission spectra of plants constitutively expressing luciferase isoforms

To generate in vivo emission spectra of all constitutively expressed luciferases, seeds were sterilized and sown on MS plates as described before (*Duan et al., 2013*). After 8 days, seedlings were treated with a 100 μM luciferin solution, incubated at room temperature for 3 hr, and imaged using an IVIS Spectrum imaging system (Perkin Elmer, Waltham, MA) using 20-nm band-pass emission filters at the following wavelengths (in nm: 490–510, 510–530, 530–550, 550–570, 570–590, 590–610, 610–630, 630–650, 650–670, 670–690, 690–710). Raw images were analyzed using Fiji and in vivo emission spectra were constructed. The full emission spectra of LUX and nanoLUC could not be constructed since the maximum of these two luciferases is below the lower band-pass filter that was available.

### Imaging system

We designed a custom-imaging system (GLO1) optimized for imaging dual-reporter luciferase expression in our custom rhizotrons. The design was a joint effort with Bioimaging Solutions (San Diego, CA), which also built the system and wrote the acquisition software that drives all the mechanical parts of the system. The system is composed by two 2048 × 2048 PIXIS-XB cameras (Princeton Instruments, Trenton, NJ) mounted on top of each other to capture two fields of view encompassing approximately two 15 × 15-cm areas corresponding to the top or bottom of the rhizotron. The cameras are fitted with a Carl-Zeiss macro lens. A filter wheel with space for four, 76.2-mm filters is positioned in front of the cameras and controlled by a stepper motor allowing for automated changing of the filter wheel position. We used two 542/50 and 450/70 custom cut Brightline(R) band-pass filters (Semrock, Rochester, NY). In single color imaging mode, the filter wheel is operated without filters. Positioned in front of the filter

wheel is a removable rhizotron holder mounted on a stepper motor. This stepper motor is also controlled by the GLO-1 software allowing automatic acquisition of images from both sides of the rhizotron sequentially. The whole-imaging system is enclosed in a light-tight black box with a door that allows loading and un-loading of rhizotrons.

## Plant imaging

Around 50 ml of 300 μM D-luciferin (Biosynth, Itasca, IL) was added to soil at the top of the rhizotron. In general, 5-min exposures were taken per rhizotron, per side, per channel. For daily imaging experiments, plants were imaged at dawn (±1 hr) to reduce possible effects on diurnal rhythms of keeping plants in the dark during imaging. Shoot images were taken using a Nikon D3100 camera.

## Image preparation

Four individual images are collected: top front, bottom front, top back, and bottom back. Using an automated ImageJ macro, a composite image is generated as follows: (1) to correct for differences in background values between the two cameras, the mean background value of each image is subtracted from 200; (2) images are rotated and translated to control for small misalignments between the two cameras; (3) the top and bottom images of each side are merged; (4) the back image is flipped horizontally; (5) the front and back images are combined using the maximum values. When dual color images are acquired, this operation is repeated for each channel. The final images produced are 16-bit in depth and 4096 × 2048 pixels. The scale of the images is 138.6 pixels per cm. Considering that an *Arabidopsis* root tip is 100 μm, this results in 1.39 pixels across an *Arabidopsis* root.

## GLO-RIA ImageJ plug-in

GLO-RIA uses a combination of existing tools to extract relevant root architecture features. Directionality is acquired using the directionality plugin from ImageJ (http://fiji.sc/Directionality). After the number of direction bins (we usually use bins of 2°) is defined by the user, a 5 × 5 Sobel operator is used to derive the local gradient orientation. This orientation is then used to build a distribution of directions by assigning the square of the orientation into the appropriate bin. Instead of representing the total counts at each orientation, a relative value is calculated by dividing the individual values at each bin by the total sum of the histogram (and multiplying by 100). Similar algorithms have been used to quantify dynamic changes in the plant cytoskeleton (*Lindeboom et al., 2013*).

The elliptic Fourier descriptors are aquired using the Fourier Shape Analysis plugin (http://imagejdocu.tudor.lu/doku.php?id=plugin:analysis:fourier_shape_analysis:start) on the convex hull shape of the root system. Elliptic Fourier descriptors have been used in numerous studies to analyze variations in shapes, notably in leaves (e.g., *Chitwood et al., 2014*, *Iwata and Ukai, 2002*).

The shape analysis is inspired by RootScape (*Ristova et al., 2013*). Due to the absence of fixed, recognizable structures in root system (that are required for the position of true landmarks), pseudo-landmarks are automatically extracted from the root systems. Shortly, the image is divided vertically at equidistant positions (with the number defined by the user) and for each of the image stripes, the minimum and maximum × coordinates are computed. The shape analysis is therefore able to discriminate root system with different vertical root distributions or global root system orientation (e.g., chemotropism). The code source for the plugin, manual, and sample images can be found in the GitHub repository of the project (https://github.com/rr-lab/GLO-Roots/tree/master/gloria).

Statistical analysis was performed in *R Developement Core Team (2014)*. The tidyr (*Wickham, 2014*), dplyr (*Wickham, 2014*), gridExtra (*Auguie, 2012*), shapes (*Dryden, 2013*), geomorph (*Adams and Otarola-Castillo, 2013*), ggplot2 (*Wickham, 2009*), and cowplot (*Wilke, 2015*) packages were used for data preparation, analysis, and plotting. Final figure preparation was done in Inkscape (https://inkscape.org/en/).

## Data availability

All the scripts and original data used to analyze and produce the images can be accessed in the GitHub repository of the project: github.com/rr-lab/GLO-Roots. Raw files of all the images used in the paper are available in Dryad (*Rellán-Álvarez et al., 2015*), http://dx.doi.org/10.5061/dryad.7tk51 (*Rellán-Álvarez et al., 2015*).

## Acknowledgements

Work in the lab of JRD was funded by the Carnegie Institution for Science Endowment and grants from the National Science Foundation (MCB-115795) and Department of Energy, Biological and Environmental Research program (DE-SC0008769). RRA was supported by a Carnegie Postdoc Fellowship and currently by Conacyt Ciencia Básica Joven Investigador grant number (CB-2014-01-238101). GL was supported by the Belgian Fonds de la Recherche Scientifique. JM was funded by the National Science Foundation (IOS-0820854). CH is funded by MGH Toteston & Fund for Medical Discovery Fellowship grant 2014A051303 and NIH R37 grant GM48707 and NSF grant MCB-0519898 awarded to Frederick Ausubel, and previously by the Gordon and Betty Moore Foundation through Grant GBMF 2550.01 from the Life Sciences Research Foundation. JV was funded by the Office of Biological and Environmental Research, Office of Science, US Department of Energy, interagency agreements DE-SC0001526 and DE-AI02-07ER64452. We thank Robert Mittler and Philip Benfey for providing seeds of ZAT12:LUC and DR5:LUC+, respectively. We thank Stacey Harmer and Mike Covington for providing the eDR5:LUC2 starting vector, used in the tomato transgenics.

We also thank Neil Robbins and members of the Dinneny lab for critical review of the manuscript and suggestions during the development of the project. We greatly appreciate Tim Doyle at the Stanford Small Animal Imaging Facility for providing advice in the use of luciferase-based imaging approaches and Heather Cartwright for advice on imaging methods and instrumentation at the Carnegie Institution for Science Imaging Facility. We thank Marlo Dreissigacker Kohn at the Stanford Product Realization Lab Room 36 for advice during the design of the rhizotron.

## Additional information

### Funding

| Funder | Grant reference | Author |
|---|---|---|
| National Science Foundation | MCB-115795 | José R Dinneny |
| U.S. Department of Energy | DE-SC0008769 | José R Dinneny |
| National Science Foundation | MCB-0519898 | Cara H Haney |
| National Science Foundation | IOS-0820854 | Amanda Schrager-Lavelle, Julin Maloof |
| National Institutes of Health | GM48707 | Cara H Haney |
| Fonds De La Recherche Scientifique - FNRS (Belgian National Fund for Scientific Research) | | Guillaume Lobet |
| Consejo Nacional de Ciencia y Tecnología (National Council of Science and Technology, Mexico) | CB-2014-01-238101 | Rubén Rellán-Álvarez |
| U.S. Department of Energy | DE-AI02-07ER64452 | John P Vogel |

The funders had no role in study design, data collection and interpretation, or the decision to submit the work for publication.

### Author contributions

RR-Á, JRD, Conception, design and development of the growth and imaging system and Arabidopsis transgenic lines; acquisition, analysis and interpretation of data; drafting and revising the article; GL, Development of the GLO-RIA image analysis plugin, analysis and interpretation of data, drafting and revising the article; HL, Acquisition of data, development of the tomato growth and imaging setup; P-LP, Acquisition of data, analysis and interpretation of data; JS, Development of Brachypodium transgenic lines, acquisition and analysis of Brachypodium, tomato and Arabidosis data; M-CY, Development of Arabidopsis and Brachypodium transgenic lines; YG, Development of Arabidopsis transgenic lines; CT, TLR, Acquisition and analysis of the QPCR data; AS-L, JM, Contributed the unpublished dual-color tomato line; CHH, Contributed the unpublished *Pseudomonas fluorescens* CH267-lux strain; RN, JPV, Contribution to the development of the Brachypodium transgenic line

Author ORCIDs
Rubén Rellán-Álvarez, http://orcid.org/0000-0001-6843-3716
Guillaume Lobet, http://orcid.org/0000-0002-5883-4572
Heike Lindner, http://orcid.org/0000-0002-5913-9803
Pierre-Luc Pradier, http://orcid.org/0000-0002-2426-0742
Jose Sebastian, http://orcid.org/0000-0002-1826-0308
Muh-Ching Yee, http://orcid.org/0000-0002-0445-7927
Charlotte Trontin, http://orcid.org/0000-0002-2235-4390
Therese LaRue, http://orcid.org/0000-0002-6805-5118
Amanda Schrager-Lavelle, http://orcid.org/0000-0003-2435-9336
Cara H Haney, http://orcid.org/0000-0002-6099-6677
Julin Maloof, http://orcid.org/0000-0002-9623-2599
John P Vogel, http://orcid.org/0000-0003-1786-2689
José R Dinneny, http://orcid.org/0000-0002-3998-724X

## Additional files

### Supplementary files

• Supplementary file 1. Blueprints of the holders, clear sheets, and spacers needed to build the rhizotrons. Additional details are provided in the 'Materials and methods'. Files are provided in Adobe Illustrator .ai and Autocad .dxf formats.

• Supplementary file 2. Primers used in the qPCR experiment.

• Supplementary file 3. Vector maps of all the constructs used in this work.

### Major dataset

The following dataset was generated:

| Author(s) | Year | Dataset title | Dataset ID and/or URL | Database, license, and accessibility information |
|---|---|---|---|---|
| Rellán-Álvarez R, Lobet G, Lindner H, Pradier PL, Sebastian J, Yee M, Geng Y, Trontin C, LaRue T, Schrager-Lavelle A, Haney CH, Nieu R, Maloof J, Vogel JP, Dinneny JR | 2015 | Data from: GLO-Roots: an imaging platform enabling multidimensional characterization of soil-grown roots systems | http://dx.doi.org/10.5061/dryad.7tk51 | Available at Dryad Digital Repository under a CC0 Public Domain Dedication. |

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
