## [Decision Letter]

Thank you for sending your work entitled “Multidimensional mapping of root responses to environmental cues using a luminescence-based imaging system” for consideration at *eLife*. Your Tools and Resources article has been evaluated by Detlef Weigel (Senior Editor) and three reviewers, one of whom is a member of our Board of Reviewing Editors.

The Reviewing Editor and the other reviewers discussed their comments before reaching this decision, and the Reviewing Editor has assembled the following comments to help you prepare a revised submission.

The reviewers agree that the system is new and exciting, and potentially offers new opportunities for imaging roots growing in soils. However, the agreement was that the data do not yet adequately demonstrate the fundamental soundness and technical details of the system. Additionally, the inclusion of many brief examples of potential biological applications has resulted in an overly long manuscript and in several instances, the biological data are apparently derived from measurements of single plants which leads to questions about the rigor of the experimentation.

When revising the manuscript, your goal should be to generate a focused manuscript that describes a rigorously-tested system and includes a smaller selection of biological applications to illustrate and highlight the potential of GLO1. Significant revision will be required.

Please note, in principle, the reviewers are excited about GLO1 but it is possible that enthusiasm for the paper will decrease when key issues such as the dynamic range of the luminescence detection, the spatial resolution of the extracted structures, and the degree of automation/throughput achievable with the platform are clarified.

1) The authors over-emphasize the differences between GLO1 and methods widely used in *Arabidopsis* research. This theme of contrast with gel-based methods, which dominates the overly-long Introduction and reemerges regularly gives the manuscript a forced-defensive feel. This should be modified. The emphasis should be on how GLO1 works and what it can measure.

2) The emphasis on the differences between GLO1 and other *Arabidopsis* growth methods also makes the authors draw some misguided interpretations. For example, in Figure 1, a large amount of gene expression data was obtained to determine if plants grown in a GLO1 setup were in a similar or different ‘state’ compared to plants grown in gels or pots. PCA shows results from plants grown in GLO1 and pots group together, and separately from the two gel-based experiments (which included MS or 25% MS salts). The authors are technically justified to conclude that GLO1 plants are more like pot-grown plants than like gel-grown plants, and they are free to view one as more ‘natural’ than the other. But the data more naturally support a different narrative. Within-treatment variation (variation between biological reps) is far greater than differences between gel and soil. (The spread of PC1, which explains 75% of the variance, is from -10 to 10 regardless of growth method or whether the samples were harvested at night or day.) Separation in PC2 supports the point the authors choose to make, which could be due entirely to MS salts being in the gels but not the soil. (MS contains 40 mM potassium nitrate, so it is not surprising that gene expression differs from soil-grown plants.) The authors should focus on explaining their method and its special features well. The readers will realize GLO1 is interesting and it is different but they will want to know how it works, how reliable and reproducible are the results, and what special studies it enables. Removing panels D-I from Figure 1 and panel A from Figure 2 could make an effective introduction to the method with one figure.

3) The spatial resolution of the images is low, approximately 72 microns per pixel. A root is therefore 1-2 pixels wide as the authors note. This raises questions. First, how much wider is the luminescence signal? From the images shown, it is not possible to determine how far it spreads, but it must spread because it is unlikely that a single-pixel signal would be detectable in the images shown. If the luminescence signal is typically 5 pixels wide or 10, can parallel roots be distinguished? It is essential to include a rigorous assessment of the resolution. Is the resolution sufficient to images of microbes or to distinguish lateral root primordia?

4) Technical details concerning registration of the images from the two cameras are very unclear and must be provided. In the subsection “Image Preparation”, it is stated that images from the two cameras must be registered (though that standard image analysis term is not used) in order to create a continuous view. The steps taken sound like an affine transformation was applied, but it is also possible that a person is manually doing the shifting by eye. A one-pixel error is equivalent to shifting the image the distance of a root thickness. Is it true that each time point for each rhizotron requires manual registration of the images from two cameras? If not manual, how accurate is the registration method? If manual, how long does it take to process a single experiment?

5) The majority of the figures in the manuscript show nice images of roots (or videos of root growth) and changes in architecture can be seen. However, relatively little quantitative data is extracted from these images and in most figures, only root angle or depth/length ratios are shown. Is the software capable of extracting additional parameters typically used to describe root architecture? The full capabilities should be described and limitations should be acknowledged. Similarly, can differences in intensity of the luciferase signal corresponding to differences in promoter activity be resolved? What is the dynamic range?

6) The measure of root system “directionality” based on a Sobel filter is interesting but not adequately described. The y-axis of the plots in Figure 3 is labeled “counts”. What is being counted and how can there be fewer than one of them? It would be expected that the area under the curves (frequency histograms?) would increase with time as more and more elements or segments detected by the Sobel operator were counted, but instead the area appears constant over time. This is one example of the lack of replication in the biological examples. The average results shown appear to be from one time series of one plant rather than multiple individual plants.

In Figure 4, it is not possible to tell whether the directionality results are the averages of 9-12 plants or only apply to the representative images shown. Can average directionality be computed? The directionality measure may be a valuable feature of the GLO1 images if appropriately described, rigorously tested and if average directionality is computed. If not, then the software is doing only very basic analyses.

7) The biological examples should focus on those that show the unique features of GLO1 and in each case, the experiments should be replicated. In Figure 7, the salt stress data appear to be derived from a single plant over time. These results should be replicated. In Figure 10, it is not clear that the gray scale intensity/water content calibration process can be applied to an image containing a root. The lightness attributed to dry pixels could be coming directly or indirectly from optical effects of the root. It is an interesting idea, but many more controls are needed. If these cannot be added, it would be better to exclude this example.

8) The title poorly matches this methods paper. The heading “GLO1: a semi-automated luminescence imaging system for rhizotrons” would be a much better title. Or, “GLO1: a luminescence imaging system for semi-automated analysis of plant root system form and gene expression”.

9) Overall, the text should be modified to make it more concise and accurate. For example, the section “GLO1: a semi-automated luminescence imaging system for rhizotrons” uses “depth” and “thick” to mean the same thing. It could be rewritten as “Apparently, the soil sheet is thick enough to block portions of the root system but thin enough to ensure its continuous structure could be compiled from opposite face views”. Additionally, please avoid making global statements that are not entirely accurate (for example, “Current image analysis algorithms are optimized for roots that are continuously visible, since they are designed to work with images of roots grown in transparent media or on paper”). There are some excellent tools for measuring roots in soil.

[Editors' note: further revisions were requested prior to acceptance, as described below.]

Thank you for submitting your work entitled “GLO-Roots: an imaging platform enabling multidimensional characterization of soil-grown roots systems” for peer review at *eLife*. Your submission has been favorably evaluated by Detlef Weigel (Senior Editor), a Reviewing Editor, and 2 reviewers.

The reviewers agree that the revisions have improved the manuscript, however, new experimental methods and software require solid validation and this is still lacking. It is essential to validate the data generated by GLO-RIA to a known ground truth, for example, a 3D printed model of a root system with known architecture. Alternatively, comparisons to manual measurements of the root system (root growth, directionality) or to measurements obtained through another independent approach could be used to validate data from GLO-RIA. The following papers provide examples of validation approaches (Topp et al., PNAS, 2013, 110(18): E1695-E1704, http://www.plantmethods.com/content/11/1/33 and http://www.plantmethods.com/content/11/1/26/figure/F6).

---

## [Author Response]

*1) The authors over-emphasize the differences between GLO1 and methods widely used in* Arabidopsis *research. This theme of contrast with gel-based methods, which dominates the overly-long Introduction and reemerges regularly gives the manuscript a forced-defensive feel. This should be modified. The emphasis should be on how GLO1 works and what it can measure*.

We have modified the text as suggested. The Introduction has been significantly reduced and now emphasizes the biological rationale for the development of GLO-Roots. The additional data included related to technical specifications of the system (see below) also directly address the specific concern of the reviewer.

*2) The emphasis on the differences between GLO1 and other* Arabidopsis *growth methods also makes the authors draw some misguided interpretations. For example, in*
Figure 1*, a large amount of gene expression data was obtained to determine if plants grown in a GLO1 setup were in a similar or different ‘state’ compared to plants grown in gels or pots. PCA shows results from plants grown in GLO1 and pots group together, and separately from the two gel-based experiments (which included MS or 25% MS salts). The authors are technically justified to conclude that GLO1 plants are more like pot-grown plants than like gel-grown plants, and they are free to view one as more ‘natural’ than the other. But the data more naturally support a different narrative. Within-treatment variation (variation between biological reps) is far greater than differences between gel and soil. (The spread of PC1, which explains 75% of the variance, is from -10 to 10 regardless of growth method or whether the samples were harvested at night or day.) Separation in PC2 supports the point the authors choose to make, which could be due entirely to MS salts being in the gels but not the soil. (MS contains 40 mM potassium nitrate, so it is not surprising that gene expression differs from soil-grown plants.) The authors should focus on explaining their method and its special features well. The readers will realize GLO1 is interesting and it is different but they will want to know how it works, how reliable and reproducible are the results, and what special studies it enables. Removing panels D-I from*
Figure 1
*and panel A from*
Figure 2
*could make an effective introduction to the method with one figure*.

We agree that deleting panels D-I from Figure 1 and combining it with Figure 2 gives a better introduction to the system and therefore we have changed the figure according to the reviewers’ suggestion.

Regarding the PCA analysis of the gene expression patterns across the different growth systems, we have found that this analysis was done with non-normalized expression data (raw CT values, not dCT values). After repeating the analysis using data normalized with housekeeping genes (dCT values) we observe a very clear separation in PC1 between plants grown in soil (rhizotrons and pots) and plants grown in MS. This first component explains 42% of the total variance. We apologize for making this error in the initial analysis. With the current analysis, we believe it is especially clear that substantial differences in gene expression exist for soil and gel-grown root systems. We have tried to emphasize in the text the main point of this experiment, which is to show that aspects of root biology that are of interest to the field at large show distinct differences between experimental conditions and that GLO-Roots provide a more pot-like environment.

*3) The spatial resolution of the images is low, approximately 72 microns per pixel. A root is therefore 1-2 pixels wide as the authors note. This raises questions. First, how much wider is the luminescence signal? From the images shown, it is not possible to determine how far it spreads, but it must spread because it is unlikely that a single-pixel signal would be detectable in the images shown. If the luminescence signal is typically 5 pixels wide or 10, can parallel roots be distinguished? It is essential to include a rigorous assessment of the resolution*. *Is the resolution sufficient to images of microbes or to distinguish lateral root primordia?*

When roots are grown in plates, and therefore both the bright field and the luminescent image can be acquired and the diameter of a root easily compared, we observe a 15% spread of the signal. Assuming that a typical root is around 140 µm, then a typical root would be around 2 pixels wide and the signal would spread another 0.30 µm, making for a total width of around 2.3 pixels. In adult root systems grown in rhizotrons, we observe an average root width (using the luminescence signal) of around 3.2 pixels. So assuming a similar root width in adult root system and in young seedlings this would suggest a signal spread of around 1 extra pixel. This would mean that the system would be able to distinguish two individual roots separated by around 70 um.

At the current focal distance, the resolution is enough to distinguish lateral root primordia. See, for example new Figure 6.

The resolution is not sufficient to resolve individual bacteria, which are about 1 micron in length; if an individual bacterium were bright enough, it would appear as a single bright pixel. Since the soil was treated with bacteria and the entire soil isn't luminescent, it is more likely that we are not imaging single bacteria but rather colonies or biofilms consisting of many bacteria.

4) Technical details concerning registration of the images from the two cameras are very unclear and must be provided. In the subsection “Image Preparation”, it is stated that images from the two cameras must be registered (though that standard image analysis term is not used) in order to create a continuous view. The steps taken sound like an affine transformation was applied, but it is also possible that a person is manually doing the shifting by eye. A one-pixel error is equivalent to shifting the image the distance of a root thickness. Is it true that each time point for each rhizotron requires manual registration of the images from two cameras? If not manual, how accurate is the registration method? If manual, how long does it take to process a single experiment?

Registration and combination of the front, back, top, and bottom images is done automatically via an ImageJ macro as a preliminary step for image analysis. The macro automatically flips, rotates and translates the different images according to precise measurements previously obtained manually in the system. Since the relative distances of the two cameras and the rhizotron holder do not change, these operations can be automatically applied to all the images. We did not included the macro in the original version, but it is now included in the Github repository:

https://github.com/rr-lab/GLO-Roots/blob/master/gloria/gloroot_combine.ijm

*5) The majority of the figures in the manuscript show nice images of roots (or videos of root growth) and changes in architecture can be seen. However, relatively little quantitative data is extracted from these images and in most figures, only root angle or depth/length ratios are shown. Is the software capable of extracting additional parameters typically used to describe root architecture? The full capabilities should be described and limitations should be acknowledged*.

Yes, there are additional data that can be extracted from the images. We chose to show the basic ones but others were included in the Github repository. A table (Table 2), describing the extracted variables, has been added to the manuscript.

With regard to directionality, GLO-RIA allows automatic partition of the image in quadrants (or rectangles) enabling the comparison of the directionality at different depths, or regions of the root system. A good example of this capability was originally shown in Figure 9—figure supplement 1 (now Figure 6—figure supplement 4). All this data is already included in the Github repository. See for example results for Figure 3: https://github.com/rr-lab/GLO-Roots/blob/master/figures/figure_3/data/high-root-data-dir.csv

It is also worth noting that in the Root Reporter plugin, a different approach is used to calculate directionality. Here, once the different segments of the root system are identified, a direction value is obtained based on the vector traced between the origin of the segment and the end point. This allows pairing of segment direction data with soil moisture values or gene expression data.

Based on the root shape defined when the root area is calculated, different types of shape descriptors can be quantified. GLO-RIA can automatically quantify Elliptic Fourier Descriptors, example here: https://github.com/rr-lab/GLO-Roots/blob/master/figures/figure_3/data/high-root-data-efd.csv. It can also extract any number of pseudo-landmarks defined by the user. Both datasets can be used for further shape and morphometric analyses, depending on the user’s needs and requirements. We did not further analyze these data, but we now show root shape analysis in Figure 3 of the new version of the manuscript.

The main limitation of GLO-RIA is its inability to reconstruct the topological links between the roots due to interruptions in root segments caused by soil particles. A statement of this limitation has been added to the main text.

Similarly, can differences in intensity of the luciferase signal corresponding to differences in promoter activity be resolved? What is the dynamic range?

Changes observed in luciferase intensity will depend on the changes in the transcript abundance driven by the promoter used. In our system, we typically observe higher expression of *UBQ10:LUC2* in the root tips. On a typical root system of a plant expressing different reporters, we can observe gray intensity values ranging from 500 to 8,000. The cameras we use are able to capture up to 64,000 gray levels, so we are well within the range to acquire even more intense signals. We have quantified the pattern of expression along the first 0.5 cm of root tips of plants expressing various reporters (UBQ10, ZAT12 and DR5), to show differences in patterns of expression and overall intensity values. This is now shown in Figure 4—figure supplement 1. DR5 expression is higher in the root tip and its range of intensity values ranges between 200 and 600 (dynamic range = 3), while ZAT12 and UBQ10 shows intensity values between 200 and 8000 (dynamic range = 40) and intensity is more spread along the root. These data and a description of the dynamics range measured has been added to the manuscript.

*6) The measure of root system “directionality” based on a Sobel filter is interesting but not adequately described. The y-axis of the plots in*
Figure 3
*is labeled “counts”. What is being counted and how can there be fewer than one of them? It would be expected that the area under the curves (frequency histograms?) would increase with time as more and more elements or segments detected by the Sobel operator were counted, but instead the area appears constant over time. This is one example of the lack of replication in the biological examples. The average results shown appear to be from one time series of one plant rather than multiple individual plants*.

Directionality is calculated as follows:

After the number of direction bins (we normally use bins of 2°) is defined by the user, a 5x5 sobel operator is used to derive the local gradient orientation. This orientation is then used to build the histogram by assigning the square of the orientation into the appropriate bin. Instead of representing the total counts at each orientation, a relative value is calculated by dividing the individual values at each bin by the total sum of the histogram (and multiplying by 100). This is why the area under the curve does not change with increasing number of segments in older roots.

For the sake of clarity, only one plant is shown in Figure 3, but the results shown in panels C and D are the average of a representative experiment with three plants. This was not indicated in the figure caption, but we have amended this. Raw images for this experiment were uploaded to Dryad and the original data was included in the Github repository (and now associated with the figure).

*In*
Figure 4*, it is not possible to tell whether the directionality results are the averages of 9-12 plants or only apply to the representative images shown. Can average directionality be computed? The directionality measure may be a valuable feature of the GLO1 images if appropriately described, rigorously tested and if average directionality is computed. If not, then the software is doing only very basic analyses*.

Yes, the values shown are the average of 9-12 plants per accession (raw images of all the plants were uploaded to Dryad) and the average can be easily computed. The original data for this image is provided in the Github repository and now associated with the image.

*7) The biological examples should focus on those that show the unique features of GLO1 and in each case, the experiments should be replicated. In*
Figure 7*, the salt stress data appear to be derived from a single plant over time. These results should be replicated*.

We have removed the time-series data for brevity and we only show the initial time point before doing the salt addition.

*In Figure 10, it is not clear that the gray scale intensity/water content calibration process can be applied to an image containing a root. The lightness attributed to dry pixels could be coming directly or indirectly from optical effects of the root. It is an interesting idea, but many more controls are needed. If these cannot be added, it would be better to exclude this example*.

To test the possibility that roots could be contributing to the lightness of the soil, we have compared the gray values of soil with no roots with those with roots. To do this we sampled several parts of different rhizotrons under both well watered and water deficit conditions. The results show no differences in gray level intensity between parts of the rhizotrons with roots or without roots. This result is now described in the text and included as Figure 5—figure supplement 2.

8) The title poorly matches this methods paper. The heading “GLO1: a semi-automated luminescence imaging system for rhizotrons” would be a much better title. Or, “GLO1: a luminescence imaging system for semi-automated analysis of plant root system form and gene expression”.

We thank the reviewers for their suggestion, however, we think the following title may be more appropriate:

“GLO-Roots: an imaging platform enabling multidimensional characterization of soil-grown roots systems”.

*9) Overall, the text should be modified to make it more concise and accurate. For example, the section “GLO1: a semi-automated luminescence imaging system for rhizotrons” uses “depth” and “thick” to mean the same thing. It could be rewritten as “Apparently, the soil sheet is thick enough to block portions of the root system but thin enough to ensure its continuous structure could be compiled from opposite face views”*.

We have changed the text accordingly in the revised manuscript to follow this recommendation.

*Additionally, please avoid making global statements that are not entirely accurate (for example, “Current image analysis algorithms are optimized for roots that are continuously visible, since they are designed to work with images of roots grown in transparent media or on paper”). There are some excellent tools for measuring roots in soil*.

We have deleted this sentence in the new version.

[Editors' note: further revisions were requested prior to acceptance, as described below.]

*The reviewers agree that the revisions have improved the manuscript, however, new experimental methods and software require solid validation and this is still lacking. It is essential to validate the data generated by GLO-RIA to a known ground truth, for example, a 3D printed model of a root system with known architecture. Alternatively, comparisons to manual measurements of the root system (root growth, directionality) or to measurements obtained through another independent approach could be used to validate data from GLO-RIA. The following papers provide examples of validation approaches (Topp et al., PNAS, 2013, 110(18): E1695-E1704*, http://www.plantmethods.com/content/11/1/33
*and*
http://www.plantmethods.com/content/11/1/26/figure/F6*).*

We agree that validation of GLO-RIA would benefit from the analysis of ground truthed images. We now include additional GLO-RIA validation using two approaches. First, we manually quantified individual lateral root angles (measurements performed as marked in blue in the diagram in Figure 9) in a set of 15 images (not originally part of the data set used in the manuscript) and calculated an average lateral root angle per root system. This value was compared with mean directionality values obtained using GLO-RIA. We also compared the manually measured root system width and depth with the same values obtained using GLO-RIA. To further extend the comparison of actual lateral root angles with the output of the directionality algorithm in GLO-RIA, we used ArchiSimple (38) to generate a set of 1240 contrasting root systems that we used as ground truth models. Images of these root systems were then measured in GLO-RIA and compared to the actual root angles defined by ArchiSimple.

Author response image 1.**DOI:**
http://dx.doi.org/10.7554/eLife.07597.059

As you can see from the results presented in Figure 1—figure supplement 5, we observed strong correlations for all three parameters tested (root system width, depth and directionality) between the manual quantification and the GLO-RIA quantification as well as between the computationally generated root models with known root angles and the GLO-RIA quantification.

We also now include examples of directionality color-coded root systems as measured by GLO-RIA for both a real root system and an ArchiSimple generated root model.

The validation provides strong evidence that the image analysis methods developed in our manuscript are accurate in their estimation of important root parameters, in which we have characterized environmental and genetic variation.